# Coordinated recruitment of Spir actin nucleators and myosin V motors to Rab11 vesicle membranes

**Olena Pylypenko[1]\*[†], Tobias Welz[2][†], Janine Tittel[3], Martin Kollmar[4], Florian Chardon[1], Gilles Malherbe[1], Sabine Weiss[2], Carina Ida Luise Michel[2], Annette Samol-Wolf[2], Andreas Till Grasskamp[2], Alistair Hume[5], Bruno Goud[1], Bruno Baron[6,7], Patrick England[6,7], Margaret A Titus[8], Petra Schwille[3], Thomas Weidemann[3], Anne Houdusse[1]\*[‡], Eugen Kerkhoff[2]\*[‡]**

[1]Institut Curie, PSL Research University, CNRS, UMR 144, F-75005, Paris, France; [2]University Hospital Regensburg, Regensburg, Germany; [3]Max Planck Institute of Biochemistry, Martinsried, Germany; [4]Max Planck Institute for Biophysical Chemistry, Göttingen, Germany; [5]University of Nottingham, Nottingham, United Kingdom; [6]Institut Pasteur, Biophysics of Macromolecules and their Interactions, Paris, France; [7]CNRS, UMR 3528, Paris, France; [8]Department of Genetics, Cell Biology and Development, University of Minnesota, Minneapolis, United States

**\*For correspondence:** Olena. Pylypenko@curie.fr (OP); Anne. Houdusse@curie.fr (AHo); eugen. kerkhoff@ukr.de (EK)

[†]These authors contributed equally to this work
[‡]These authors also contributed equally to this work

**Competing interests:** The authors declare that no competing interests exist.

**Abstract** There is growing evidence for a coupling of actin assembly and myosin motor activity in cells. However, mechanisms for recruitment of actin nucleators and motors on specific membrane compartments remain unclear. Here we report how Spir actin nucleators and myosin V motors coordinate their specific membrane recruitment. The myosin V globular tail domain (MyoV-GTD) interacts directly with an evolutionarily conserved Spir sequence motif. We determined crystal structures of MyoVa-GTD bound either to the Spir-2 motif or to Rab11 and show that a Spir-2:MyoVa:Rab11 complex can form. The ternary complex architecture explains how Rab11 vesicles support coordinated F-actin nucleation and myosin force generation for vesicle transport and tethering. New insights are also provided into how myosin activation can be coupled with the generation of actin tracks. Since MyoV binds several Rab GTPases, synchronized nucleator and motor targeting could provide a common mechanism to control force generation and motility in different cellular processes.

## Introduction

Cytoskeletal filaments provide tracks for intracellular transport, an essential function for the organization, polarity and communication of eukaryotic cells. Polarized microtubules span the cytoplasm and serve as tracks for fast long-range transport mediated by dynein and kinesin motors. Myosin molecular motors have a functional area limited by the space in which F-actin tracks are found in cells. The very dynamic nature of the actin cytoskeleton promotes transport on more flexible routes beyond the microtubule network. Recent advances in understanding of the functional links between cargos, myosins and actin assembly regulators demonstrate that they cooperate to accomplish specific membrane transport events (*Cheng et al., 2012*; *Schuh, 2011*; *Sirotkin et al., 2005*; *Sun et al., 2006*). For example, the cooperation of class I myosin with the actin nucleator Arp2/3 complex is required for membrane remodeling in endocytosis (*Cheng et al., 2012*; *Sirotkin et al., 2005*; *Sun et al., 2006*). In mouse oocytes, actin nucleators Spir and formin-2 (FMN2) cooperate to generate actin filaments at Rab11 vesicle membranes, while myosin Vb (MyoVb) mediates the Rab11

vesicle long-range transport towards the oocyte cortex at metaphase (*Pfender et al., 2011*; *Schuh, 2011*). Moreover, nucleus positioning in mouse oocytes results from myosin V driven active diffusion of Rab11 vesicles, which also requires Spir:formin-2 dependent actin polymerization on these membranes (*Ahmed et al., 2015*; *Almonacid et al., 2015*).

However, the mechanism by which motors and actin nucleators are both specifically recruited to membranes is poorly understood at present. Direct interaction between the membrane anchored active Rab11 small GTPase and MyoV (MyoVa and MyoVb) globular tail domain (GTD) is necessary to target the motor to Rab11 vesicles (*Lapierre et al., 2001*; *Lindsay et al., 2013*). In contrast, a direct interaction between Spir and Rab11 could not be detected and it is still unclear how differential recruitment of the motors and the actin track regulators can be specified to promote Rab11-vesicle transport. The Spir actin nucleators (Spir-1 and Spir-2 in mammals) encode a FYVE-type zinc finger membrane-binding domain at their C-termini (*Kerkhoff et al., 2001*; *Otto et al., 2000*; *Tittel et al., 2015*) that promiscuously interacts with negatively charged lipids (*Tittel et al., 2015*). It has been proposed that the interaction of Spir proteins with additional factors provides the specificity for its targeting to the correct subpopulation of vesicles (*Tittel et al., 2015*).

A search for the mechanism that controls the targeting of the Spir:FMN complex to Rab11:MyoV vesicles revealed a conserved MyoV globular tail domain binding motif (GTBM) in Spir proteins that mediates a direct interaction between Spir and MyoV. We have solved the crystal structures of the MyoVa globular tail domain in complex with Spir-2-GTBM and in complex with Rab11. We have also shown that the MyoV-GTD links Spir-2 and Rab11 into a tripartite complex *in vitro* and in cells. The Spir:MyoV interaction contributes to the motor activation and to the coordination of the specific membrane recruitment of both actin polymerization and motor machineries required for force production powered by MyoV.

## Results

### Spir directly interacts with myosin V

To gain insights into whether the actin nucleator Spir and the motor MyoV could be both activated in coordination on vesicle membranes, we first performed protein interaction studies to determine if Spir and MyoV (*Figure 1*) coexist in a protein complex. Our initial GST-pulldown experiments showed that GST-MyoVb-GTD is able to pull endogenous Spir-1 from mouse brain lysates, as does the GST-FMN2-eFSI protein that binds directly to the Spir KIND domain (*Pechlivanis et al., 2009*) as a positive control (*Figure 2—figure supplement 1A*). In co-immunoprecipitation (co-IP) experiments employing human embryonic kidney cells transiently over-expressing recombinant Spir and MyoV, the full-length Spir-1 and Spir-2 proteins did interact with the GFP-MyoVb-GTD (*Figure 2A*). We mapped the Spir sequences necessary for the interaction with MyoVb by successive N-terminal deletions of Spir-2. The deletion of KIND and WH2 domains did not affect complex formation; however, further deletion of the linker region between the WH2 domains and the Spir-box (*Figure 1*) completely impaired the interaction (*Figure 2A*), demonstrating that the Spir central linker region is important for MyoVb-GTD binding. The GTDs of MyoVb and MyoVa are highly conserved (*Pylypenko et al., 2013*) and both directly interact with Rab11 (*Lindsay et al., 2013*; *Roland et al., 2009*). Interestingly, GFP-MyoVa-GTD also interacts with Spir-2 (*Figure 2—figure supplement 1B*). This is consistent with the fact that MyoVa and MyoVb have overlapping cellular functions (such as mobilization of Rab11 recycling endosomes for the AMPA receptor transport into dendritic spines [*Correia et al., 2008*; *Hammer and Wagner, 2013*; *Wang et al., 2008*]), share interacting partners and utilize similar mechanisms for Rab11-vesicle transport.

We confirmed a direct interaction of the two MyoV isoforms (MyoVa, MyoVb) and Spir by GST-pulldown assays using purified recombinant proteins (*Figure 2B*). Further experiments showed that both MyoVa and MyoVb bind the Spir-2-linker (aa 361–519) with similar affinities in the sub-micromolar range ($K_d \sim 728$ nM and 377 nM for MyoVa and MyoVb, respectively) (*Figure 2C*).

### Evidence for Spir:MyoV complex formation at vesicle membranes

The detected tight binding of MyoV and Spir suggested that they would interact in cells. Transgenic co-expression of tagged full-length MyoVa (eGFP-MyoVa-FL) and Spir-2 (Myc-Spir-2-FL) proteins in HeLa cells showed a perfect colocalization of the two proteins on vesicular structures in the central

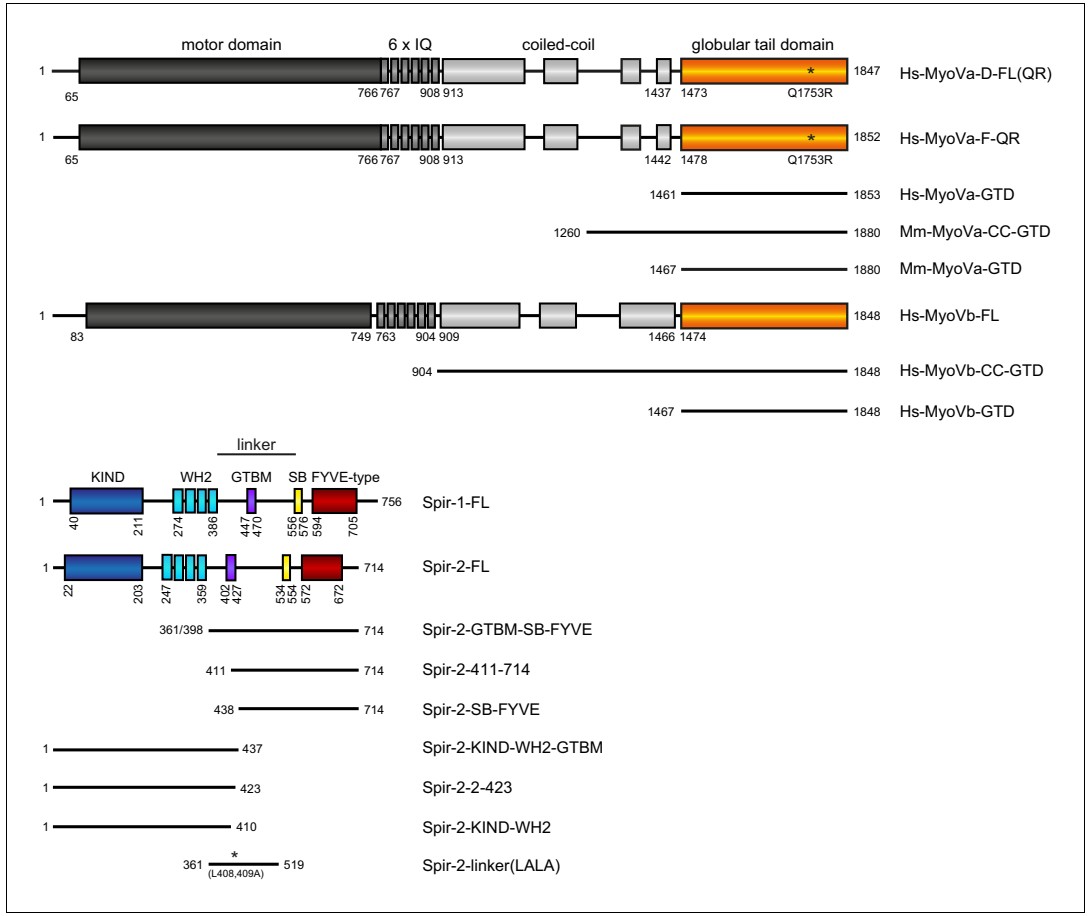

**Figure 1.** Schematic overview of the Spir and MyoV protein fragments used in this study. The myosin V motor domain and its 6 IQ lever arm is followed by a coiled-coil dimerization region and the C-terminal globular tail domain. The central linker region of Spir connects the N-terminal KIND and four actin binding WH2 domains on one side, with the C-terminal Spir-box (SB) and a membrane binding FYVE-type zinc finger on the other. The newly identified Spir myosin V binding motif (GTBM) is located in the middle of the linker region. The domain boundaries are indicated in the full-length MyoVa (containing exon D (D) or exon F (F)), MyoVb, Spir-1 and Spir-2 proteins. Numbers indicate amino acids. Stars indicate amino acid substitutions. *FL*, full-length; *GTD*, globular tail domain; *CC*, coiled-coil; *KIND*, kinase non-catalytic C-lobe domain; *GTBM*, globular tail domain binding motif; *SB*, Spir-box; *LALA*, L408,409A substitution. Species abbreviations are Hs, *Homo sapiens*; Mm, *Mus musculus*.

cytoplasm, consistent with formation of a Spir:MyoV complex at vesicle membranes (*Figure 3A*). This was also the case for MyoVb and Spir-2 (*Figure 3B*). In order to confirm a complex formation of Spir-2 and MyoVa on vesicle membranes we performed FLIM-FRET (fluorescence life time imaging, fluorescence resonance energy transfer) microscopy experiments (*Sun et al., 2013*) with co-expressed AcGFP-MyoVa-GTD as a donor fluorophore and mStrawberry-Spir-2 deletion mutants as acceptor fluorophores. A mStrawberry-tagged C-terminal Spir-2 protein (mStrawberry-Spir-2-GTBM-SB-FYVE), including the Spir-GTBM (necessary for MyoV interaction), the Spir-box and the FYVE-type zinc finger (Spir membrane interaction) is able to reduce the fluorescence lifetime of the AcGFP-MyoVa-GTD donor at vesicle membrane surfaces (*Figure 3C–E*). In contrast, a mStrawberry-fusion protein encoding only the Spir-box and the FYVE-type zinc finger (mStrawberry-Spir-2-SB-FYVE) or a mutant Spir-2-GTBM-SB-FYVE-LALA protein, in which two essential leucines for MyoV interaction were replaced by alanines, did not alter the lifetime of the AcGFP-MyoVa-GTD donor (*Figure 3E*). As the lifetime reduction indicates that MyoVa and Spir-2 proteins are in close proximity (within 10 nm of distance between the fluorophores), these results provide a strong support for a direct interaction of the two proteins at vesicle membranes. The MyoVa-GTD is monomeric in

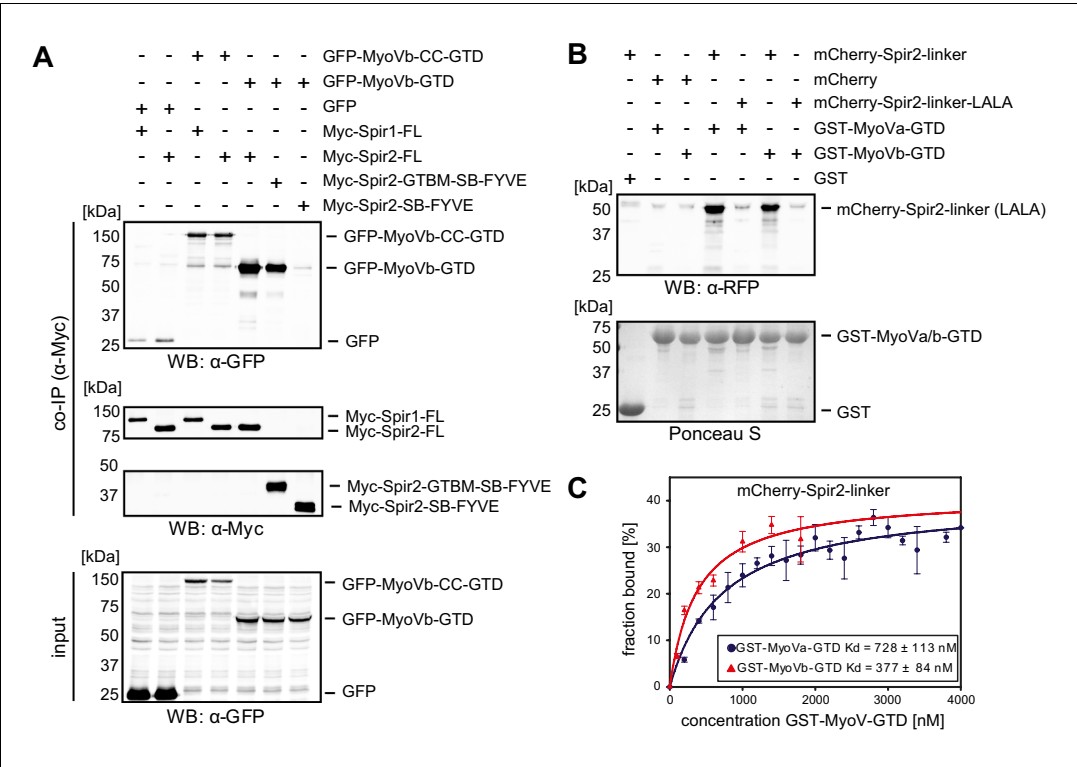

**Figure 2.** Spir proteins directly interact with myosin V. (**A**) Co-immunoprecipitation assay of HEK293 cells transfected with plasmids expressing AcGFP (GFP), AcGFP-tagged MyoVb deletion mutants and indicated Myc-epitope tagged Spir-1 and Spir-2 proteins. Cell lysates were immunoprecipitated with anti-Myc antibodies. The cell lysates (input) and immunoprecipitates (co-IP) were analyzed by immunoblotting with antibodies as indicated. The Spir-2-GTBM and the MyoVb globular tail domain (GTD) were identified as being essential for the Spir/MyoV interaction. N = 4 experimental repeats. *WB*, Western blotting. (**B**) GST-pulldown studies employing purified bacterially expressed recombinant proteins to analyze a direct interaction of Spir-2-linker and MyoVa/Vb-GTD. The mutation of two highly conserved leucines within the Spir-2-GTBM to alanines (human Spir-2-L408A, L409A; Spir-2-linker-LALA, see also *Figure 4A*) largely impairs binding of this mutant to GST-MyoVa/Vb-GTD. N = 4 experimental repeats. (**C**) Fluorescence spectroscopy was used to determine the dissociation constants ($K_d$) for MyoVa and MyoVb GTD binding to His$_6$-mCherry-Spir-2-linker. *Error bars* represent SEM (n = 4 experimental repeats).

The following figure supplement is available for figure 2:

**Figure supplement 1.** Interaction of endogenous Spir-1 with MyoVb-GTD.

solution (*Figure 5—figure supplement 2*). To exclude a FLIM-FRET signal by a dense packing of proteins at vesicle membranes, we co-expressed green and red-fluorescent protein tagged MyoVa-GTDs as a donor/acceptor pair (AcGFP-MyoVa-GTD, mStrawberry-MyoVa-GTD) which revealed a similar FRET efficiency as the donor alone (*Figure 3E*), indicating that a direct interaction of MyoVa-GTD and the Spir-2-GTBM is required for the observed FLIM-FRET signal.

## Identification of a conserved MyoV binding motif in Spir

The linker regions of vertebrate Spir-1 and Spir-2 proteins show low overall sequence homology (for example similarity 21%, identity 11% between human Spir-1 and Spir-2 linker regions) (*Figure 4—figure supplement 1*) but a short sequence of 27 amino acids (*Figure 4A*) in the middle part of the linker is more conserved (similarity 56%, identity 37%) suggesting that the fragment could possibly have an important role in the recruitment of MyoV. Interestingly, Spir-2 fragments containing the fully conserved regions interact with MyoVb-GTD, whereas no interactions were observed with fragments lacking half of this region (*Figure 4—figure supplement 2*). Mutation of two highly conserved

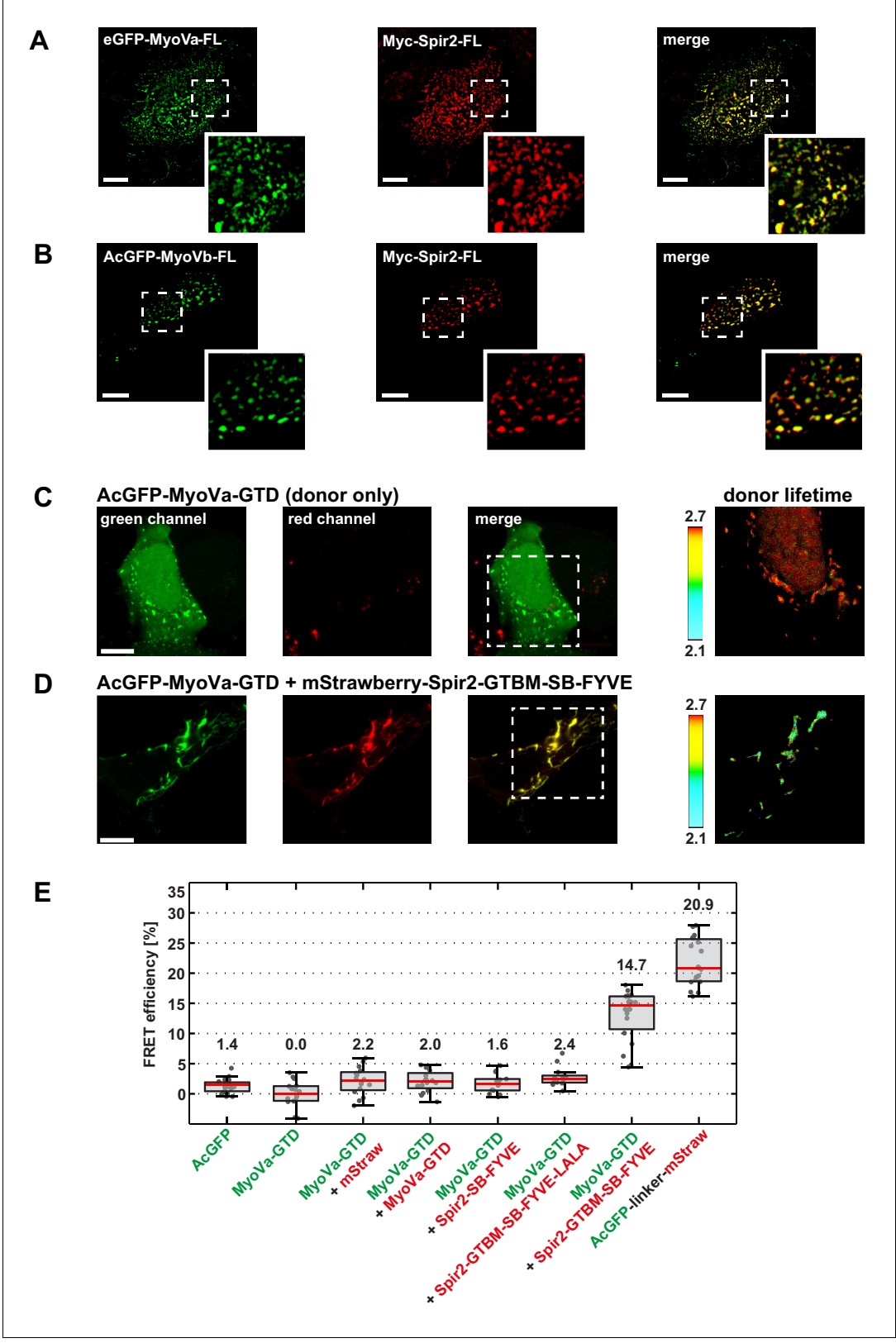

**Figure 3.** Myosin V and Spir-2 interact at vesicle membranes. (**A** and **B**) GFP-tagged full-length MyoVa (eGFP-MyoVa-FL; green; **A**) and MyoVb (AcGFP-MyoVb-FL; green; **B**) colocalize with Myc-epitope-tagged full-length Spir-2 (Myc-Spir-2; red) at vesicle membranes when transiently co-expressed in HeLa cells as indicated by overlapping punctae (merge; yellow; higher magnification in insets). 5 cells were recorded and one representative cell is

*Figure 3 continued on next page*

*Figure 3 continued*

presented here. *Scale bars* represent 10 μm. (**C–E**) FLIM-FRET analysis of transiently expressed AcGFP-tagged MyoVa-GTD (AcGFP-MyoVa-GTD, donor) and mStrawberry-tagged C-terminal Spir proteins (acceptors) at vesicle membranes in HeLa cells. (**C, D**) Examples demonstrating the lifetime shift due to FRET. Confocal fluorescence images (green channel, AcGFP; red channel, mStrawberry; and fluorescence lifetime images of AcGFP) of AcGFP-MyoVa-GTD expressed alone (**C**) and in the presence of the interacting acceptor protein mStrawberry-Spir-2-GTBM-SB-FYVE (**D**) are shown. (**E**) The average FRET efficiencies per cell measured at vesicle membranes for the indicated transiently expressed donor-acceptor combinations are presented in a box-and-whisker plot. Every dot represents a single cell. The region of interest was manually confined to the cytoplasm (average FRET efficiencies for AcGFP alone and the tandem AcGFP-linker-mStrawberry). For all other experiments, the ROI was further reduced by a threshold algorithm that identifies vesicles in the AcGFP channel. Box-and-whisker plots indicate 2nd and 3rd quartile (box), median (red horizontal line, value noted above each box), and 1.5x interquartile range (whiskers). 10–15 cells have been analyzed for each transfection.

The following source data is available for figure 3:

**Source data 1.** Source data for FLIM-FRET analysis of MyoVa and Spir-2 expression at vesicle membranes.

Spir-2 residues within this newly identified Spir linker homology region, Leu408Ala and Leu409Ala, is sufficient to abolish direct binding to MyoV-GTD (*Figure 2B*). The identified minimal MyoV globular tail domain binding motif of Spir (Spir-GTBM), corresponding to the human Spir-2 short peptide (aa 401–427) binds MyoVa and MyoVb GTDs with low micromolar affinity (*Figure 4B*).

## Spir binding site is conserved in all three MyoV isoforms

We further characterized the Spir:MyoVa binding by solving a crystal structure of the complex, which allowed us to identify the protein interaction sites and depict the stabilizing interactions within the complex at atomic detail (*Figure 4C*, *Table 1*). Importantly, the Spir residues important for the interaction with MyoV-GTD are conserved among Spir proteins (*Figure 4A*). The MyoV-GTD is composed of two closely apposed subdomains (SD-1 and SD-2) (*Figure 4C*). The Spir-2-GTBM peptide binds to the SD-1 of the MyoVa-GTD in an extended conformation, forming a two-turn alpha helix at the end, in the cleft between helices H3 and H5 (*Figure 4C,D*). Spir-2-GTBM can be accommodated on the surface of MyoVa without conformational changes in the GTD (*Figure 4—figure supplement 3A,B*), and the Spir-GTBM binding pocket is structurally conserved in MyoVb and MyoVc (*Figure 4—figure supplement 3C,D*). Consistently, a direct binding assay also showed that MyoVc-GTD binds Spir-2-GTBM with micromolar range affinity (*Figure 4—figure supplement 3E*). We thus conclude that the Spir-binding site structural conservation of the three myosin V isoforms explains Spir promiscuity in binding to MyoVa, Vb and Vc.

## Spir and melanophilin bind to the same MyoVa pocket, but use different interaction modes

Interestingly, the MyoV-GTD Spir-2 binding site partially overlaps with that of melanophilin (MLPH) (*Pylypenko et al., 2013*; *Wei et al., 2013*) (*Figure 4D,E*). Both Spir-2-GTBM and MLPH-GTBM bind the MyoVa-GTD with similar micromolar range affinities (*Figure 4B*) (*Pylypenko et al., 2013*). However, in contrast to the Spir-GTBM, which interacts with all three MyoV isoforms, the MLPH-GTBM binding is MyoVa specific (*Wei et al., 2013*; *Pylypenko et al., 2013*). The Spir-2 and MLPH GTBMs exhibit some sequence similarity (*Figure 4F*) and comparison of the complex structures shows that their N-terminal regions interact with MyoVa in a very similar way, forming a network of hydrogen bonds with the H4''-H5-loop and similarly positioning two conserved leucine residues of the motif (L408 and L409 in Spir-2; L188 and L189 in MLPH) within cavities of the MyoVa-GTD surface (*Figure 4D,E,H*). As mentioned above, mutation of the two highly conserved Spir leucine residues to alanines significantly impairs Spir-2 binding to MyoV (*Figure 2B*). The Spir-GTBM C-terminal part anchors itself in the MyoVa hydrophobic cleft between helices H3 and H5 using L414 and M417 side chains; and the small side chain of the strictly conserved Ala411 binds in proximity to the MyoVa Tyr1596 side chain (*Figure 4D,G*). The apo-MyoV-GTD surface is preformed to bind Spir-GTBM (*Figure 4G,H*). In contrast to Spir-GTBM, the MLPH-GTBM cannot be accommodated on the surface

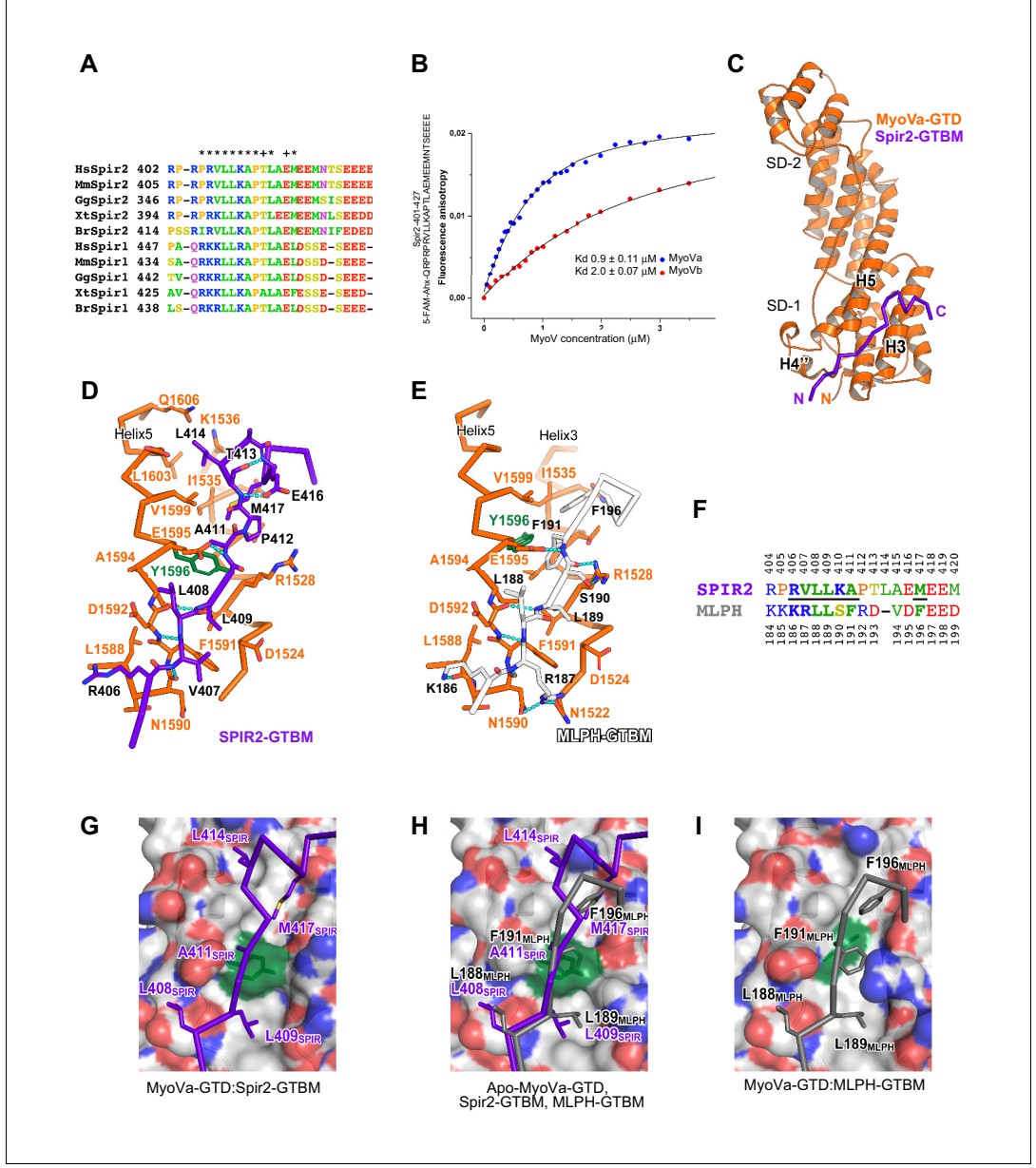

**Figure 4.** A conserved sequence motif of Spir binds to MyoV-GTD. (**A**) A short highly conserved sequence motif within the central Spir linker region is responsible for myosin V binding. Alignment of vertebrate Spir-1 and Spir-2 sequence fragments. The short sequence motif of about 27 amino acids in the middle part of the Spir linker region shows high sequence homology (see also *Figure 4—figure supplement 1*). Species abbreviations: *Homo sapiens* (Hs), *Mus musculus* (Mm), *Gallus gallus* (Gg), *Xenopus tropicalis* (Xt), *Brachydanio rerio* (Br). All sequence-related data are available through CyMoBase (http://www.cymobase.org). Hs-Spir-2 residues contacting MyoVa within 4 Å distance are labeled with '*'; the Spir-2 conformation is stabilized by two conserved residues (see panel D) indicated with '+'. (**B**) Fluorescence anisotropy measurements of the binding of the Fluorescein-Spir-2-GTBM peptide (human, amino acid residues 401–427) to the MyoVa and MyoVb GTDs. The equilibrium dissociation constants ($K_d$) for MyoVa and MyoVb GTDs were determined by fitting the titration curves as detailed in the Materials and Methods section. The experiments were repeated twice with two different protein preparations. (**C**) Crystal structure of the MyoVa-GTD:Spir-2-GTBM complex. Spir-2-GTBM (purple) binds to subdomain-1 (SD-1) of MyoVa-GTD (orange). (**D**) Close-up view of the Spir-2-GTBM bound to MyoVa-GTD. Residues forming the interaction sites are shown as sticks and are labeled. Spir-2 E416 and T413 form intramolecular hydrogen bonds (dashed lines). Spir-2 V407, L409 and MyoVa N1590 and D1592 backbone atoms form an intermolecular β-structure like hydrogen bond network. (**E**) MLPH-GTBM bound to MyoVa-GTD (PDB ID 4LX2). The N-terminal part of MLPH-GTBM (residues from K186 to L189) interacts with MyoVa in a similar manner to Spir-2 (residues 406–409). A similar

*Figure 4 continued on next page*

*Figure 4 continued*

hydrogen bond between MyoVa E1595 and the main chain nitrogen of MLPH F191 is also observed in the Spir-2: MyoVa interface. (**F**) Structure based sequence alignment of the Spir-2-GTBM and MLPH-GTBM fragments. Residues making similar contacts with MyoVa are highlighted with a black line. (**G**) Hydrophobic residues anchoring Spir-2-GTBM on the MyoVa-GTD surface. Spir-2 A411 is packed on the top of MyoVA Y1596 (green) side chain (see also panel D). (**H**) Spir-2 (purple) and MLPH (gray) GTBMs docked on the surface of apo-MyoVa-GTD (PDB ID 4LX1). The apo-MyoVa-GTD hydrophobic cleft between the H5 and H3 is compatible with Spir-2 binding, but not with MLPH, where the side chain of F191 clashes with MyoVa Y1596 (green). In Spir, the conserved L414 side chain anchors the C-terminal Spir-2 fragment extending the interacting hydrophobic surface compared to what is found for MLPH-GTBM binding. (**I**) Hydrophobic residues anchoring MLPH-GTBM in the MyoVa-GTD pocket (PDB ID 4LX2). The MyoVa Y1596 (green) side chain is rotated to bury the side chain within the protein core (see also panel E) to accommodate MLPH F191 in the binding pocket.

The following figure supplements are available for figure 4:

**Figure supplement 1.** Gene and protein sequence structures of vertebrate Spir.

**Figure supplement 2.** Mapping of the Spir-2 myosin binding domain.

**Figure supplement 3.** Comparison of myosin V isoforms Spir/MLPH binding sites.

**Figure supplement 4.** MyoVa conformational change upon MLPH binding.

**Figure supplement 5.** Structural similarities of Spir and melanophilin.

of apo-MyoVa-GTD without conformational changes (*Figure 4H*). MLPH-GTBM has a conserved hydrophobic residue (Phe, Ile or Val) Phe191 at the position equivalent to the Spir-2-A411 (*Figure 4E,F,H*) and its binding to MyoVa requires rotation of the Tyr1596 side chain in towards the protein core to harbor the big Phe side chain in the binding pocket (*Figure 4H,I*; *Figure 4—figure supplement 4*). As proposed previously (*Pylypenko et al., 2013*), the conformational change required for MLPH binding is likely more difficult to achieve in other MyoV isoforms due to the sequence differences in the protein core surrounding the Tyr1596. Moreover, the MyoVa Arg1528 that stabilizes the MLPH-GTBM by two hydrogen bonds (*Figure 4E*) is not conserved in the myosin V isoforms and contributes to the MyoVa binding specificity (*Wei et al., 2013*). The observed differences in how Spir-2-GTBM and MLPH-GTBM bind to MyoV-GTD thus account for their discrepancy in binding specificity.

## Common features of GTBM

The globular tail domain of MyoV thus binds different partners using the same binding site. Interestingly, Spir-GTBM and MLPH-GTBM have a similar distribution of charged residues along the sequence: their N-terminal part is positively charged, and their C-terminal sequence, after the specific binding motif, contains a cluster of negatively charged residues (*Figure 4—figure supplement 5*). Thus, Spir-2-GTBM binding to MyoV also likely use the charge complementarity described for MLPH-GTBM binding to MyoVa (*Pylypenko et al., 2013*). The low amino acid complexity within the GTBM makes it very difficult to generate a reasonable binding motif profile that can be used for a search of other GTBM containing proteins against protein sequence databases. Unfortunately, our attempts to find other proteins potentially capable of MyoV binding failed. However, the polypeptide chain recognized by the GTD has three characteristic properties as follows: (1) the clusters of charged residues may help to orient the peptide relative to the GTD surface, (2) the extended N-terminal region is able to form hydrogen bonds with backbone atoms of the GTD, and (3) the hydrophobic residues of the motif provide the peptide anchoring in small surface pockets of the GTD.

In summary, we identified a new functional region in Spir and provide an atomic description of how Spir proteins interact with the GTD of MyoV motors, which brings critical insights into the compatibility or competition of Spir binding with different MyoV cargos and their ability to regulate MyoV activity.

**Table 1.** X-ray diffraction data collection and structure refinement statistics.

|  | MyoVa-GTD:Rab11 | MyoVa-GTD:Spir2 |
|---|---|---|
| **Data collection** |  |  |
| X-ray source | SOLEIL PX1 | SOLEIL PX1 |
| Space group | C 1 2 1 | C 1 2 1 |
| Cell dimensions<br>a, b, c [Å]<br>α, β, γ [°] | 215.79, 128.42, 89.02<br>90, 98.27, 90 | 99.61, 41.22, 108.22<br>90, 115.89, 90 |
| Resolution [Å] | 48.88–2.056<br>(2.18–2.056) | 44.81–1.76<br>(1.86–1.76) |
| $R_{sym}$ | 5.3 (55.5) | 7.4 (60.0) |
| I / σI | 20.6 (3.38) | 15.7 (2.5) |
| Completeness [%] | 98.59 (97.0) | 99.77 (99.75) |
| Redundancy | 7.7 (7.6) | 6.4 (6.2) |
| **Refinement** |  |  |
| Resolution [Å] | 48.88–2.056<br>(2.129–2.056) | 44.81–1.8<br>(1.864–1.8) |
| No. of reflections | 146774 (14323) | 36967 (3659) |
| Rwork / Rfree | 17.97 / 20.83<br>(23.00 / 26.6) | 15.13 / 19.13<br>(27.49 / 36.49) |
| Number of atoms in AU<br>Protein/ligand/solvent | 12516 / 172 / 1154 | 3191 / / 432 |
| Average B-factor | 60.5 | 31.9 |
| r.m.s.d<br>bond lengths [Å]<br>angles [°] | 0.002<br>0.59 | 0.01<br>1.092 |
| PDB ID | 5JCZ | 5JCY |

## A tripartite Spir:MyoV:Rab11 complex determines Spir membrane specificity

To gain insights into the relationship between Spir, MyoV and Rab11, we solved the crystal structure of Rab11a bound to the MyoVa-GTD. This structure confirms that Rab11a binds to the MyoVa-GTD SD-2, as previously reported for the MyoVb:Rab11a complex structure (*Pylypenko et al., 2013*) and indicates that some structural variability in Rab11a occurs upon its binding to the MyoV isoforms (*Figure 5—figure supplement 1*, *Table 1*). The structures of MyoVa:Spir-GTBM and MyoVa:Rab11a demonstrate that the Rab11 and Spir binding sites are on opposite ends of the elongated MyoV-GTD while the GTD adopts a similar conformation in the two binary complexes. This implies that the two MyoV partners can bind to the GTD at the same time, as shown in a model of the tripartite complex (*Figure 5A*). The organization of the tripartite complex, where MyoV links Spir to Rab11, provides an explanation of how Spir proteins can be specifically targeted to Rab11 vesicles.

To reveal the existence of the tripartite complex *in vitro* and *in vivo*, several experiments were designed. First, we showed that bacterially expressed constitutively active GST-Rab11a-Q70L pulls down transiently over-expressed full-length MyoVa, Spir-2 and a C-terminal FMN2 protein (mStrawberry-FMN2-FH2-FSI) from HEK293 lysates (*Figure 5B*); over-expressed MyoVa-Q1753R mutant impaired in Rab11 binding (*Lindsay et al., 2013*) affects all three proteins co-precipitation with the GST-Rab11a-Q70L-beads consistent with the role of MyoV for linking the MyoV:Spir:FMN2 complex to Rab11.

GST-pulldown experiments with recombinant proteins using GST-Spir-2-GTBM-SB-FYVE also confirm that MyoV serves as a linker between Spir and Rab11. In the absence of the MyoVa-GTD, Spir does not pull down Rab11a-Q70L (*Figure 5E*). The tripartite complex formation was further

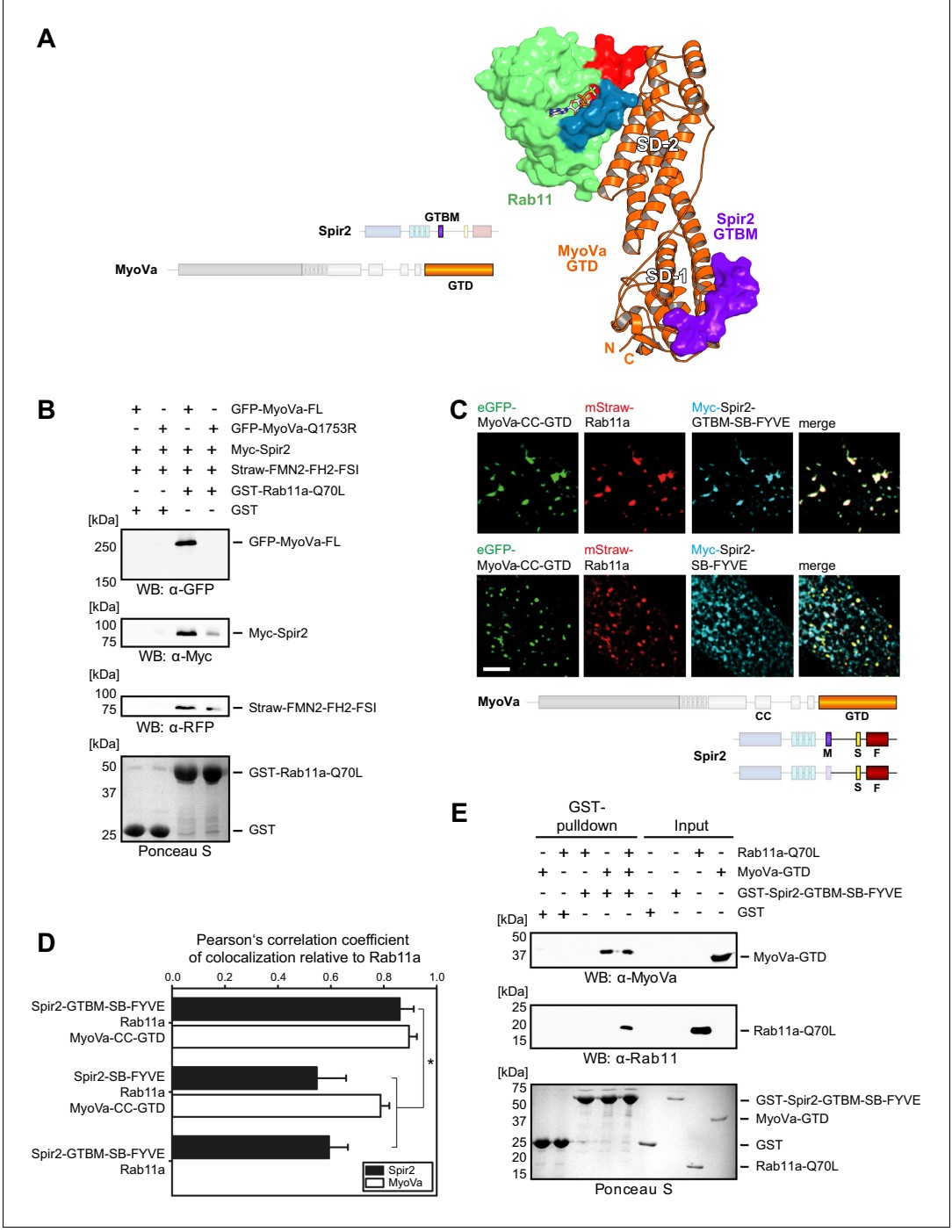

**Figure 5.** Myosin V links Spir-2 and Rab11 into a tripartite complex. (**A**) A model of the Rab11a:MyoVa-GTD:Spir-2-GTBM complex was generated by superimposition of the MyoVa-GTD from the two crystal structures Rab11a:MyoVa-GTD and MyoVa-GTD:Spir-2-GTBM. The Spir-2-GTBM (purple) binds to the SD-1 of the MyoVa-GTD (orange) and Rab11a (green, Switch-1 blue, Switch-2 red) binds to its SD-2. (**B**) GST-pulldown assay with purified GTP-locked GST-Rab11a-Q70L mutant and HEK293 cell lysates transiently over-expressing full-length Myc-epitope-tagged Spir-2 (Myc-Spir-2), mStrawberry-tagged C-terminal formin-2 (Straw-FMN2-FH2-FSI), eGFP-tagged full-length MyoVa (GFP-MyoVa-FL) or eGFP-tagged full-length MyoVa with the Q1753R mutation, which disrupts interaction with Rab11, (GFP-MyoVa-Q1753R). GST-Rab11a-Q70L is able to pull GFP-MyoVa-FL from cell lysates, but not the GFP-MyoVa-Q1753R mutant. In the presence of GFP-MyoVa-FL, GST-Rab11a-Q70L is also able to pull down Myc-Spir-2, as well as Straw-FMN2-FH2-FSI. Only faint Myc-Spir-2 and Straw-FMN2-FH2-FSI bands were detected with the GFP-MyoVa-Q1753R mutant. N = 2 experimental repeats. (**C**) The localization of transiently co-

*Figure 5 continued on next page*

*Figure 5 continued*

expressed tagged Rab11 (mStrawberry, mStraw-Rab11; red), MyoVa-CC-GTD (eGFP, eGFP-MyoVa-CC-GTD; green) and the Myc-epitope tagged (Myc, cyan) C-terminal Spir-2 proteins encoding (Myc-Spir-2-GTBM-SB-FYVE) or lacking (Myc-Spir-2-SB-FYVE) the MyoV binding motif was analyzed by fluorescence microscopy. Deconvoluted pictures indicate the localization of the proteins on vesicular structures. *Scale bars* represent 5 μm. 5 cells were recorded for each condition and the cytoplasmic region of one representative cell is presented here. (**D**) The colocalization of tagged proteins as described in (**C**) was quantified for the indicated co-expressions by determining its Pearson's correlation coefficient (PCC) as shown in a bar diagram. Each bar represents the mean PCC value for at least 4 cells analyzed. *Error bars* represent SEM. Statistical analysis was done using Student's t-test to compare two co-expression conditions with a confidence interval of 95%. *p<0.05. *Figure 5D*. (**E**) GST-pulldown assay engaging purified GST-tagged Spir-2-GTBM-SB-FYVE protein, purified MyoVa-GTD and the purified GTP-locked Rab11a-Q70L mutant. GST-Spir-2-GTBM-SB-FYVE alone does not interact with Rab11a-Q70L. In contrast, the presence of the MyoVa-GTD allows GST-Spir-2-GTBM-SB-FYVE to pull Rab11a-Q70L, as indicated by immunoblotting with antibodies recognizing MyoVa and Rab11a. N = 3 experimental repeats.

The following source data and figure supplements are available for figure 5:

**Source data 1.** Source data for calculation of mean PCC values for colocalization analysis.

**Figure supplement 1.** Rab11 binds to MyoVa-GTD at the same site as to MyoVb-GTD, but adopts a different conformation in the two complexes.

**Figure supplement 2.** The MyoVa-GTD links Rab11 and Spir-2-linker into a tripartite complex.

supported by analytical gel filtration experiments showing co-elution of all three proteins (Rab11a-Q70L, MyoVa-GTD, mCherry-Spir-2-linker) (*Figure 5—figure supplement 2*).

Fluorescence microscopy of immunostained (Myc-Spir-2-GTBM-SB-FYVE) and fluorescent proteins (eGFP-MyoVa-CC-GTD, mStrawberry-Rab11a) revealed that the three proteins have a nearly perfect colocalization when transiently co-expressed in HeLa cells (*Figure 5C,D*). The strong colocalization of Spir and Rab11 was dependent on the Spir MyoV binding motif. The Spir-2-SB-FYVE fragment, that lacks the MyoV binding motif, showed membrane localization driven by the FYVE domain (*Tittel et al., 2015*). When co-expressed with Rab11a and the MyoVa-CC-GTD, a significant reduction in Spir-2-SB-FYVE colocalization with Rab11 was observed, while colocalization of Rab11 with MyoVa-CC-GTD remained high (*Figure 5C,D*). These data strongly support that MyoV acts as an adapter to target Spir actin nucleators specifically towards Rab11 vesicle membranes.

## Spir activates Rab11 dependent MyoV membrane targeting

MLPH-GTBM binding to the MyoVa-GTD promotes a structural rearrangement in myosin Va *in vitro* whereby it switches from an inhibited 'OFF' conformation (*Figure 6A,B*) in which the GTD folds back and binds to the motor domain, to an extended active 'ON' conformation (*Yao et al., 2015*). Interestingly, the Spir and MLPH binding site also overlaps with the MyoVb-GTD N-terminal linker interaction site that has been proposed to contribute to the MyoV OFF state stabilization (*Pylypenko et al., 2013*) (*Figure 6A*). The obvious structural similarity between Spir-GTBM and MLPH-GTBM binding to the MyoVa-GTD suggests that Spir could similarly play a role in MyoV activation by disrupting the head-tail interaction and opening of the motor.

Full-length MyoVa transiently expressed in cells has an even cytoplasmic distribution (*Figure 6C*) (*Lindsay et al., 2013*), indicating that the endogenous Rab11 binding activity by itself is not sufficient to target the inhibited motor protein to vesicle membranes. To investigate if the interaction of full-length MyoVa with the Spir-GTBM contributes to the activation of the motor protein we co-expressed full-length MyoVa (eGFP-MyoVa-FL) and the N-terminal Spir-2 proteins in HeLa cells (*Figure 6D*). When expressed alone, the N-terminal Spir-2 proteins (Spir-2-KIND-WH2; Spir-2-KIND-WH2-GTBM) have an even cytoplasmic and nuclear localization (*Figure 6C*). Co-expressed MyoVa and the N-terminal Spir-2 lacking the GTBM (Myc-Spir-2-KIND-WH2) are also both cytosolic (*Figure 6D*). This is consistent with MyoV being in the inhibited state and not interacting with Spir-KIND-WH2. In contrast, when full-length MyoVa is co-expressed with a Spir-2 fragment containing the Spir-GTBM (Myc-Spir-2-KIND-WH2-GTBM), both proteins localize to vesicle membranes. This is

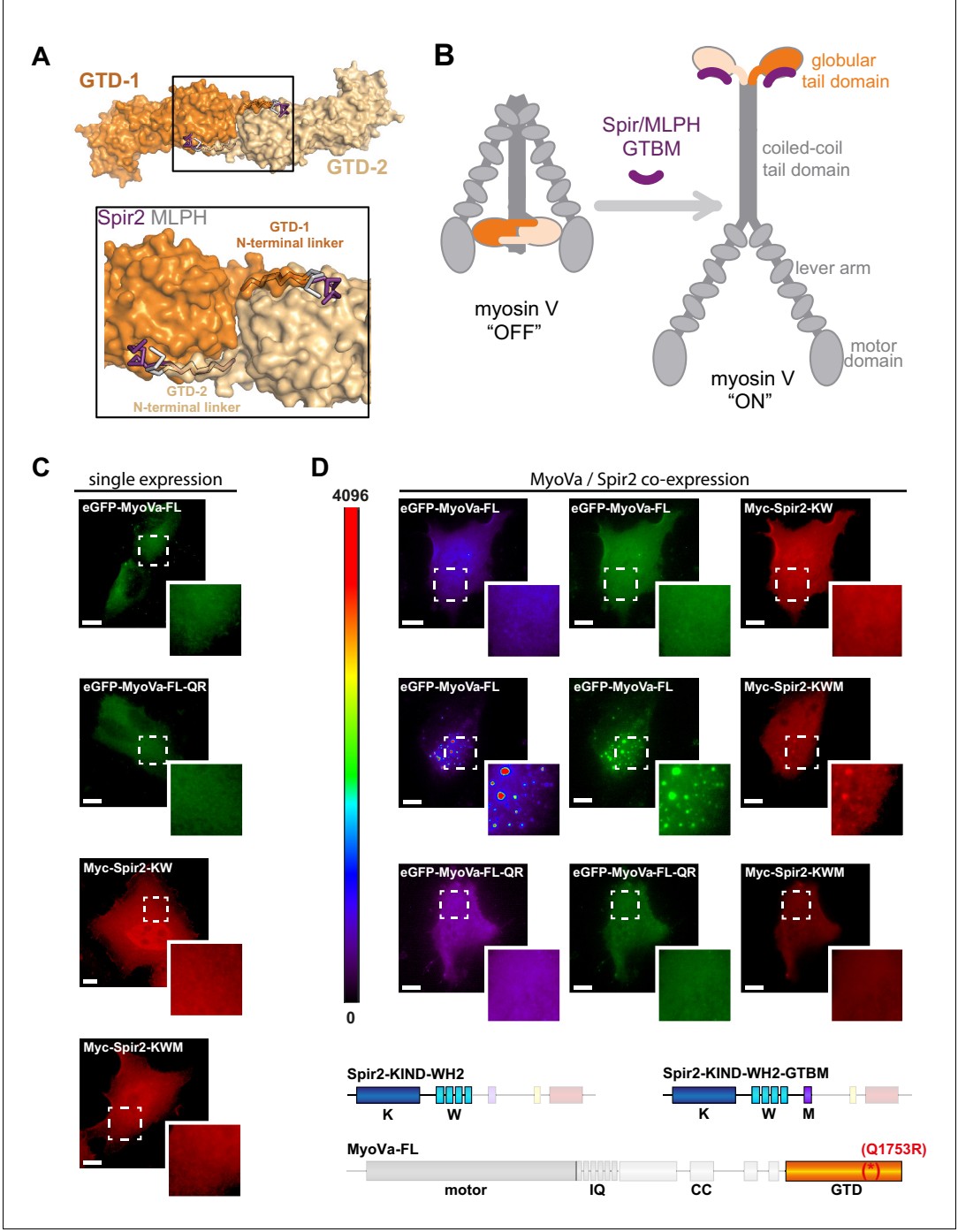

**Figure 6.** Spir-2 facilitates myosin V recruitment to vesicle membranes. (**A**) A putative interaction mode of the GTDs within the MyoV off state was derived from the MyoVb-GTD structure (PDB ID 4LX0). Two GTDs interact with each other via N-terminal linkers occupying the Spir/MLPH binding pockets on the neighboring GTD, Spir and MLPH GTBMs are shown in ribbon. (**B**) Schematic representation of myosin V activation by GTBM binding. (**C** and **D**) N-terminal Spir fragments target MyoVa to vesicle membranes. (**C**) N-terminal Spir fragments (Myc-Spir-2-KIND-WH2, Myc-Spir-2-KW; Myc-Spir-2-KIND-WH2-GTBM, Myc-Spir-2-KWM) and full-length MyoVa motor proteins (eGFP-MyoVa-FL, eGFP-MyoVa-FL-Q1753R) have an even cytoplasmic and nuclear localization when transiently expressed in HeLa cells (single expression). At least 5 cells were recorded for each condition and one representative cell is presented here. *Scale bars* represent 10 μm. (**D**) Expression of full-length GFP-tagged MyoVa (eGFP-MyoVa-FL; green (middle) and as heat map (left)) shows an even cytoplasmic distribution that is not changed by co-expression of the Spir-2 fragment (Myc-Spir-2-KIND-WH2), which is not able to bind to MyoV and

*Figure 6 continued on next page*

*Figure 6 continued*

lipid membranes (upper panel). In contrast, co-expression of a myosin V binding Spir-2 fragment (Myc-Spir-2-KIND-WH2-GTBM; red, middle panel) leads to targeting of MyoVa to vesicle membranes and to overlapping Spir-2 and MyoVa localization (higher magnification insets). Heat maps represent grey values for MyoVa fluorescence intensities rising from '0' (black) to '4096' (red) to document equal expression levels of MyoVa proteins in the depicted cells. To address Rab11 dependence on motor protein targeting, the GFP-tagged melanocyte specific F isoform of the Q1753R mutant MyoVa (eGFP-MyoVa-QR) that does not bind Rab11 was expressed. The expressed MyoVa mutant has an even cytoplasmic distribution that was not changed upon co-expression of the myosin V binding Spir-2 fragment (Myc-Spir-2-KIND-WH2-GTBM; red, lower panel). Representative cells are shown. All cells observed under single and co-expression conditions had a vesicular or cytoplasmic localization as shown by the representative cells presented here. 5 cells were recorded for each condition and one representative cell is presented here. *Scale bars* represent 10 μm.

The following figure supplement is available for figure 6:

**Figure supplement 1.** Rab11 binding may affect the GTD:motor-domain interaction in the folded inhibited state of MyoV.

consistent with a model in which Spir-GTBM binding to cytosolic MyoV activates the motor and promotes its membrane recruitment.

The vesicular localization of the full-length MyoVa triggered by Spir-2-GTBM binding was Rab11 dependent, since a MyoVa point mutant that cannot interact with Rab11 (MyoVa-Q1753R) (*Lindsay et al., 2013*) fails to be targeted to vesicle membranes and shows an even cytoplasmic distribution, when co-expressed with the Spir-2-KIND-WH2-GTBM protein (*Figure 6D*). The experiments demonstrate a Rab11-dependent coordinated membrane targeting mechanism of Spir and MyoV (*Figure 7*).

## Discussion

The timing and mechanism of motor recruitment to specific vesicles is vital for cellular function and its intracellular organization. Targeted recruitment of either dynein, kinesins or myosins to vesicles is the essential first step in sorting intracellular cargo within cells yet the logistics of motor activation and vesicle localization are poorly understood. Rab11 GTPases recruit several different molecular motors to vesicles, regulating trafficking of exocytic and recycling vesicles along microtubule and actin tracks (*Welz et al., 2014*). The actin nucleator Spir is also present on the surface of Rab11-vesicles, enabling them to generate their own tracks for transport by MyoV (*Schuh, 2011*). Here it is revealed how the activation of Spir actin nucleators and MyoV actin-based motors on Rab11 vesicles are coordinated.

The full-length MyoV motor is maintained in a folded, autoinhibited OFF state by an interaction between the N-terminal motor domain and the C-terminal GTD. Relief of this interaction by cargo or calcium binding abrogates head-tail binding, resulting in an extended, activated motor (*Li et al., 2008*; *Liu et al., 2006*; *Thirumurugan et al., 2006*) (*Figure 6B*). The MyoV motor domain and Rab11 compete for binding to the GTD since the surfaces involved in these interactions are in part overlapping (namely the H11- H12 loop; *Figure 6—figure supplement 1*). This explains why the membrane-anchored Rab11 GTPase alone is not sufficient to target the cytosolic inhibited MyoV to Rab11 positive vesicles *in vivo* (*Lindsay et al., 2013*). However, a drastic change in the localization of MyoV, from cytosol to membranes, occurs when the cellular concentration of Spir-GTBM is increased (*Figure 6D*) strongly implicating Spir in the recruitment and activation of MyoV on Rab11 vesicles.

The Spir-GTBM interacts with the same binding pocket in the MyoV-GTD SD-1 (*Figure 6A*) that can also be occupied by a short N-terminal linker preceding the MyoV-GTD implicated in stabilizing the OFF state of the myosin (*Pylypenko et al., 2013*; *Zhang et al., 2016*). Binding of the Spir-GTBM would displace the N-terminal linker and contribute to MyoV activation. MyoV activation is thus predicted to involve a competition between inter-GTDs stabilizing interactions within the motor OFF state and Spir binding to SD-1. The Spir and MLPH GTBMs share an overlapping binding site, suggesting that MLPH could activate MyoV in a manner similar to Spir. Thus, the Spir/MLPH binding site

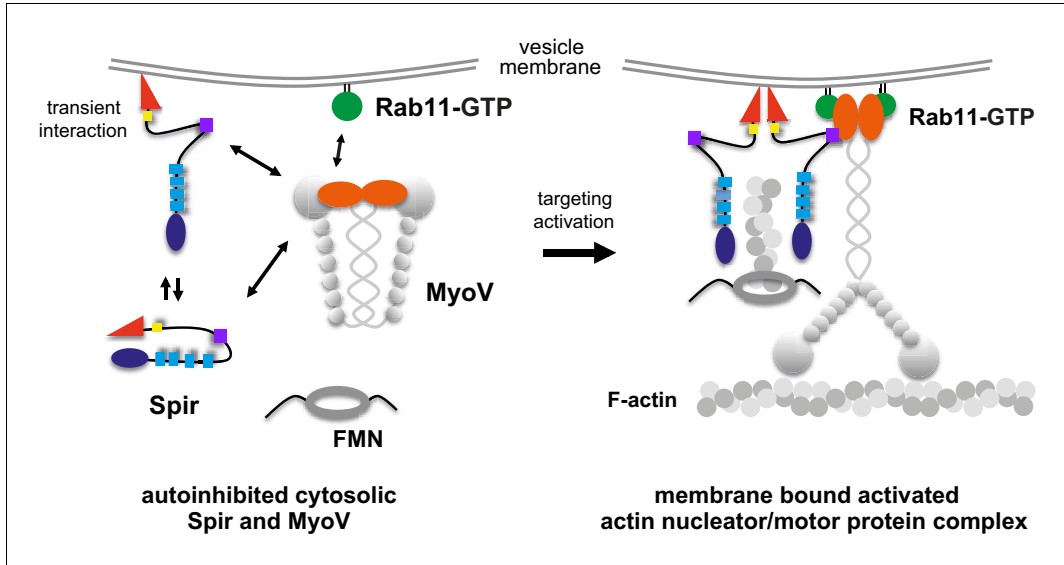

**Figure 7.** A model of a coordinated assembly of the Spir/FMN F-actin nucleator complex and myosin V motor proteins at Rab11 vesicle membranes. Spir and MyoV proteins adopt a backfolded autoinhibited conformation in the cytoplasm (*Li et al., 2008*; *Liu et al., 2006*; *Thirumurugan et al., 2006*; *Tittel et al., 2015*). A non-specific transient interaction of the Spir FYVE-type domain with membranes opens up Spir (*Tittel et al., 2015*). Spir-GTBM binding to the inhibited MyoV contributes to the release of MyoV autoinhibition and facilitates MyoV-GTD interaction with Rab11 at vesicle membranes, further stabilizing the MyoV activated extended conformation. The Spir-FYVE membrane interaction releases the cis-regulatory KIND/FYVE domain interaction and subsequently allows formin (FMN) dimer recruitment at the membranes (*Tittel et al., 2015*). Interaction with the formin dimer promotes Spir dimerization and allows efficient F-actin nucleation (*Dietrich et al., 2013*; *Namgoong et al., 2011*; *Quinlan et al., 2007*). Note that the order of events is not known and will require further studies. The domain structures of the Spir and MyoV proteins are indicated by the same color code as in *Figure 1* (Spir: FYVE, red; Spir-box, yellow; GTBM, purple; WH2, light blue; KIND, dark blue; MyoV: GTD, orange; coiled coil, calmodulin bound IQ motifs and motor domain, gray).

likely represents a MyoV activation hot spot. Further evidence and detail on the mechanism of MyoV activation by cargo awaits future structural studies on the MyoV OFF state and on other GTD-binding partner interactions.

The data presented here reveal a mechanism for how both MyoV and Spir increase their membrane colocalization and work cooperatively during vesicle transport. Spir proteins are also predicted to exist in a folded state in the cytoplasm. Promiscuous binding of the Spir-FYVE to negatively charged phospholipids is thought to promote an open conformation of the protein (*Tittel et al., 2015*), facilitating formin recruitment and assembly of an active Spir:FMN complex that promotes actin nucleation on vesicles. Together, the vesicle-bound Spir and Rab11 could then synergistically activate and recruit MyoV. The resulting membrane bound tripartite complex then coordinates nucleation of F-actin tracks and myosin force generation on a subset of Rab11 vesicles. This enables MyoV-driven motility in regions of the cell distal from the cortical actin network, such as in the oocyte cytoplasm (*Schuh, 2011*) and most likely in the central region of somatic cells. The newly synthesized actin filaments emanating from vesicles would also favor MyoV-dependent motility as this myosin has been found to have both an increased landing rate and run lengths on ADP-P$_i$ actin filaments (*Zimmermann et al., 2015*) and facilitate actomyosin mediated transport of vesicles towards microtubule tracks (*Dietrich et al., 2013*).

The exact timing and order of events required to assemble the Spir:MyoV:Rab11 tripartite complex remains to be established. Precisely defining the steps that activate and recruit MyoV and Spir to Rab11 positive vesicles will require analysis of proteins expressed at endogenous levels. The use of novel gene editing technology in combination with the next generation of fluorescent proteins (e.g. mNeonGreen) (*Shaner et al., 2013*) should enable the generation of cell lines for quantification

of endogenous protein concentration at vesicle membranes, analysis of the fluctuations of Spir, FMN, MyoV and Rab11-FIP2 at Rab11 vesicle membranes and correlation of complex dynamics with vesicular motility and the morphological dynamics of the vesicles.

The mechanism for cooperative membrane recruitment of Spir and MyoV via interaction with a Rab described here serves as a paradigm for how cells can generate actin-based transport of a range of membrane vesicles. Specifically, the insights revealed here into how Spir interacts with MyoV identify a new role for Spir as a critical regulator of MyoV recruitment and activation. Different splice forms and variants of Spir and MyoV are implicated in a wide range of cellular functions, and MyoV interacts with an array of Rab GTPases with roles in a range of transport processes (e.g. Rab3, Rab8 and Rab 32; [*Bultema et al., 2014*; *Lindsay et al., 2013*; *Roland et al., 2011*]). This suggests the possibility that Spir:MyoV complexes might be recruited via Spir and Rab membrane interactions to a number of different membrane compartments for transport along their own self-generated local actin network. For example, a splice form of Spir-1 (Spire1C) is specifically localized to the mitochondria (*Manor et al., 2015*) and it is tempting to speculate that it plays a role in recruiting MyoV that has been implicated in regulating the trafficking of this organelle (*Pathak et al., 2010*; *Schwarz, 2013*; *Sheng, 2014*). Interestingly, it has previously been shown that a class I myosin recruits the Arp2/3 actin nucleator complex to membranes during endocytic vesicle formation (*Cheng et al., 2012*). Together with the identification here of a tripartite Spir:MyoV:Rab11 complex that recruits formins to membranes, coordination of actin nucleation and myosin motors is likely to be a general strategy used by cells to promote myosin dependent motility of membranes.

## Materials and methods

### Identification and assembly of Spir sequences

The Spir genes were identified by TBLASTN (RRID: SCR_011822) searches against the respective sequenced eukaryotic genomes. The published sequences of *Drosophila* Spir and human Spir were taken as seeds. All hits were manually analyzed at the genomic DNA level by comparing the three reading frame translations to the multiple sequence alignments of all Spir proteins to reveal homologous regions missing in gene predictions. If available for a certain species, EST data from the NCBI EST database has been analyzed to help in the annotation process. All sequence-related data and references to genome sequencing centers are available through CyMoBase (http://www.cymobase.org).

### Cloning of bacterial and mammalian protein expression vectors

Prokaryotic and eukaryotic expression vectors were generated by standard cloning techniques using Pfx DNA polymerase (AccuPrime; ThermoFisher, Waltham, MA, USA), restriction endonucleases (New England Biolabs (NEB), Frankfurt am Main, Germany) and T4 DNA ligase (NEB). Point mutants were generated using the QuikChange site-directed mutagenesis kit (Agilent Technologies (former Stratagene), Santa Clara, CA, USA) and Pfu DNA polymerase (Promega, Mannheim, Germany). Sequence correctness was verified by sequencing (LGC Genomics, Berlin, Germany). A detailed overview of all expression vectors employed in this study is presented in *Table 2*.

### Recombinant protein production for structural studies and interaction measurements

Recombinant GST-MyoVa-GTD, GST-MyoVb-GTD, GST-Spir-2-GTBM-SB-FYVE, GST-Rab11a-Q70L, GST-FMN2-eFSI and His$_6$-mCherry-tagged Spir-2-linker proteins were expressed in *Escherichia coli* Rosetta or Rosetta (DE3) pLysS (Merck Millipore, Novagen, Darmstadt, Germany). Bacteria were cultured in LB medium (100 mg/l ampicillin, 30 mg/l chloramphenicol) at 37°C until an A600 nm of OD 0.6–0.8. Protein expression was induced by 0.1 mM Isopropyl-β-D-thiogalactopyranoside (IPTG; Sigma-Aldrich, Taufkirchen, Germany) and continued at 16–20°C for 18–20 hr. Bacteria were harvested and lysed by ultra-sonication. Soluble proteins were purified by an ÄKTApurifier system (GE Healthcare Life Sciences, Freiburg, Germany) using GSH-Sepharose 4B (GE Healthcare Life Sciences) or Ni-NTA beads (Qiagen, Hilden, Germany) and size exclusion chromatography (High Load 16/60 Superdex 200; GE Healthcare Life Sciences). Proteins were concentrated by ultrafiltration using

**Table 2.** Expression vectors employed in this study.

| Construct | Plasmid | Description | Fragment boundaries (restriction sites) | Purification | Purpose |
|---|---|---|---|---|---|
| Myc-Spir1-FL | pcDNA3-Myc-hs-Spir1 | Myc-hs-Spir1 | aa 2 - 757 (*Bam*HI / *Sac*I) | | Co-IP |
| Myc-Spir2-FL | pcDNA3-Myc-hs-Spir2 | Myc-hs-Spir2 | aa 2 - 714 (*Bam*HI / *Hind*III) | | Co-IP |
| Myc-Spir2-KWM | pcDNA3-Myc-hs-Spir2-KIND-WH2-GTBM | Myc-hs-Spir2-KIND-WH2-GTBM | aa 2 - 437 (*Bam*HI / *Xho*I) | | Co-IP, Transient expression |
| Myc-Spir2-2-423 | pcDNA3-Myc-hs-Spir2-2-423 | Myc-hs-Spir2-aa2-423 | aa 2 - 423 (*Bam*HI / *Xho*I) | | Co-IP |
| Myc-Spir2-KW | pcDNA3-Myc-hs-Spir2-KIND-WH2 | Myc-hs-Spir2-KIND-WH2 | aa 2 - 410 (*Bam*HI / *Xho*I) | | Co-IP, Transient expression |
| Myc-Spir2-MSF | pcDNA3-Myc-hs-Spir2-GTBM-SB-FYVE | Myc-hs-Spir2-GTBM-SB-FYVE | aa 361/398 - 728 (*Bam*HI / *Xho*I) | | Co-IP, Transient expression |
| Myc-Spir2-411-714 | pcDNA3-Myc-hs-Spir2-411-714 | Myc-hs-Spir2-aa411-714 | aa 411 - 714 (*Bam*HI / *Xho*I) | | Co-IP |
| Myc-Spir2-SF | pcDNA3-Myc-hs-Spir2-SB-FYVE | Myc-hs-Spir2-SB-FYVE | aa 438 - 714 (*Bam*HI / *Xho*I) | | Co-IP, Transient expression |
| Strawberry-Spir2-MSF | pmStrawberry-C1-hs-Spir2-GTBM-SB-FYVE | mStrawberry-hs-Spir2-GTBM-SB-FYVE | aa 361 - 714 (*Eco*RI / *Kpn*I) | | Transient expression, FLIM-FRET |
| Strawberry-Spir2-SF | pmStrawberry-C2-hs-Spir2-SB-FYVE | mStrawberry-hs-Spir2-SB-FYVE | aa 438 - 714 (*Xho*I / *Bam*HI) | | Transient expression, FLIM-FRET |
| Strawberry-Spir2-MSF-LALA | pmStrawberry-C1-hs-Spir2-GTBM-SB-FYVE-LALA | mStrawberry-hs-Spir2-GTBM-SB-FYVE-L408A,L409A | aa 361 - 714 (*Eco*RI / *Xba*I) | | Transient expression, FLIM-FRET |
| GST-Spir2-MSF | pGex-4T3-hs-Spir2-GTBM-SB-FYVE | GST-hs-Spir2-GTBM-SB-FYVE | aa 361 - 714 (*Bam*HI / *Xho*I) | GSH-Sepharose 4B, Superdex 200 | GST-Pulldown |
| His-mCherry-Spir2-linker | pProEx-HTb-mCherry-hs-Spir2-linker | His$_6$-mCherry-hs-Spir2-linker | aa 361 - 519 (*Xho*I / *Hind*III) | Ni-NTA, Superdex 200 | GST-Pulldown, Binding assays |
| His-mCherry-Spir2-linker-LALA | pProEx-HTb-mCherry-hs-Spir2-linker-LALA | His$_6$-mCherry-hs-Spir2-linker-L408A,L409A | aa 361-519 (mutagenesis) | Ni-NTA, Superdex 200 | GST-Pulldown |
| Spir2 / Fluorescein-Spir2 peptide | | | aa 401-427 | purchased from GenScript | Crystallization, Fluoresc. anisotropy |
| GFP-MyoVa-D-FL | provided by Bruno Goud | *Lindsay et al., 2013* | | | GST-Pulldown, Transient expression |
| GFP-MyoVa-D-QR | provided by Bruno Goud | *Lindsay et al., 2013* | | | GST-Pulldown, Transient expression |
| GFP-MyoVa-F-QR | provided by Bruno Goud | *Lindsay et al., 2013* | | | Transient expression |
| GFP-MyoVa-CC-GTD | peGFP-C2-mm-MyoVa-CC-GTD | eGFP-mm-MyoVa-CC-GTD | aa 1260 - 1880 (*Eco*RI / *Sal*I) | | Transient expression |
| GFP-MyoVa-GTD | pAcGFP-C1-mm-MyoVa-GTD | AcGFP-mm-MyoVa-GTD | aa 1467 - 1880 (*Hind*III / *Sal*I) | | Co-IP, FLIM-FRET |
| Strawberry-MyoVa-GTD | pmStrawberry-C1-mm-MyoVa-GTD | mStrawberry-mm-MyoVa-GTD | aa 1467 - 1880 (*Hind*III / *Sal*I) | | Transient expression, FLIM-FRET |
| GST-MyoVa-GTD | pGex-4T1-NTEV-mm-MyoVa-GTD | GST-mm-MyoVa-GTD | aa 1467 - 1880 (*Hind*III / *Sal*I) | GSH-Sepharose 4B, Superdex 200 | GST-Pulldown, Binding assays |
| MyoVa-GTD | pProEx-HTb-hs-MyoVa-GTD | hs-MyoVa-GTD | aa 1461 - 1853 (*Bam*HI / *Xho*I) | HisTrap, TEV, Superdex 200 | Crystallization, GST-Pulldown, Anisotropy |
| GFP-MyoVb-CC-GTD | pAcGFP-C1-hs-MyoVb-CC-GTD | AcGFP-hs-MyoVb-CC-GTD | aa 904 - 1848 (*Hind*III / *Sal*I) | | Co-IP |
| GFP-MyoVb-GTD | pAcGFP-C1-hs-MyoVb-GTD | AcGFP-hs-MyoVb-GTD | aa 1467 - 1848 (*Hind*III / *Sal*I) | | Co-IP, FLIM-FRET, Transient expression |
| GFP-MyoVb-GTD-Q1748R | pAcGFP-C1-hs-MyoVb-GTD-Q1748R | AcGFP-hs-MyoVb-GTD-Q1748R | aa 1467 - 1848 (mutagenesis) | | Transient expression |

*Table 2 continued on next page*

*Table 2 continued*

| Construct | Plasmid | Description | Fragment boundaries (restriction sites) | Purification | Purpose |
|---|---|---|---|---|---|
| GST-MyoVb-GTD | pGex-4T1-NTEV-hs-MyoVb-GTD | GST-hs-MyoVb-GTD | aa 1467 - 1848 (*Bam*HI / *Xho*I) | GSH-Sepharose 4B, Superdex 200 | GST-Pulldown, Binding assays |
| MyoVb-GTD | pProEx-HTb-hs-MyoVb-GTD | hs-MyoVb-GTD | aa 1456 - 1848 (*Bam*HI / *Xho*I) | HisTrap, TEV, Superdex G200 | Crystallization, GST-Pulldown, Anisotropy |
| MyoVc-GTD | pProEx-HTb-hs-MyoVc-GTD | hs-MyoVc-GTD | aa 1350 - 1742 (*Bam*HI / *Xho*I) | HisTrap, TEV, Superdex 200 | Microscale thermophoresis |
| Strawberry-Rab11a | pmStrawberry-C1-mm-Rab11a | mStrawberry-mm-Rab11a | aa 2 - 216 (*Eco*RI / *Bam*HI) | | Transient expression |
| GST-Rab11a-Q70L | pGex-4T1-NTEV-cl-Rab11a-Q70L | GST-cl-Rab11a-Q70L | aa 1 - 216 (*Eco*RI / *Eco*RI) | GSH-Sepharose 4B, Superdex 200 | GST-Pulldown |
| Rab11a | pET28-rTEV-hs-Rab11a | hs-Rab11a | aa 1 - 177 (*Nco*I / *Eco*RI) | HisTrap, TEV, Superdex 200 | Crystallization, SPR |
| Rab11a-Q70L | pET28-rTEV-hs-Rab11a-Q70L | hs-Rab11a-Q70L | aa 1 - 173 (*Nco*I / *Eco*RI) | HisTrap, TEV, Superdex 200 | GST-Pulldown |
| Strawberry-FMN2-FH2-FSI | pmStrawberry-C2-mm-FMN2-FH2-FSI | mStrawberry-mm-FMN2-FH2-FSI | aa 1135 - 1578 (*Bam*HI / *Xho*I) | | GST-Pulldown |
| GST-FMN2-eFSI | pGex-4T1-NTEV-mm-FMN2-eFSI | GST-mm-FMN2-eFSI | aa 1523 - 1578 (*Bam*HI / *Xho*I) | GSH-Sepharose FF, Superdex 200 | GST-Pulldown from brain lysates |
| GFP | pAcGFP-C1 | AcGFP | | | Co-IP, FLIM-FRET |
| GFP-linker-Strawberry | pAcGFP-C1-linker-mStrawberry | AcGFP-linker(A-S-G-A-G)-mStrawberry | aa 1 - 236 (*Bsp*EI / *Bgl*II) | | FLIM-FRET |
| mCherry | pProEx-HTb-mCherry | His₆-mCherry | aa 1 - 236 (*Bam*HI / *Xho*I) | Ni-NTA, Superdex 200 | GST-Pulldown |
| GST | pGex-4T1-NTEV | GST | | GSH-Sepharose 4B, Superdex 200 | GST-Pulldown |

Amicon Ultra centrifugal filters (Merck Millipore, Darmstadt, Germany) with respective molecular weight cut offs. The final protein purity was estimated by SDS-PAGE and Coomassie staining.

MyoVa-GTD, MyoVb-GTD, MyoVc-GTD and Rab11a-1-176 were produced and purified as previously described (*Pylypenko et al., 2013*). Briefly, the recombinant expression of MyoVb, MyoVa or MyoVc GTD domains (residues 1456–1848, 1461–1853, 1350–1742 respectively) was performed in *Escherichia coli* BL21(DE3) or BL21-CodonPlus-RILP cells using a pProEX-HTb vectors containing an N-terminal 6xHis peptide and rTEV cleavage site. The Rab11a (1–176), His₆-mCherry-Spir-2-linker and Rab11a-Q70L (1–173) were produced using a pProEX-HTb or a pET28 modified vector containing N-terminal His₆ tag and rTEV cleavage site in BL21(DE3) cells.

Bacterial cells were grown at 37°C in 2YT medium induced at an A600 nm of OD 0.6 by the addition of 1 mM IPTG, and harvested after 18 hr at 20°C. The cell pellet was resuspended in 50 mM Tris pH 8.0, 300 mM NaCl, 2 mM MgCl₂, 30 mM imidazole, 2 mM β-mercaptoethanol, 5% (v/v) glycerol and protease inhibitor mix (Chymostatin, Leupeptin, Antipain, Pepstatin A at 1 µg/ml), lysed by sonication, and centrifuged at 35,000 x *g* for 1 hr. The supernatant was loaded onto a HisTrap column (GE Healthcare Life Sciences). After washing with the buffer, the fusion proteins were eluted with a gradient of 30–500 mM imidazole. The His-tag was cleaved by incubation with rTEV protease at 1:50 molar ratio overnight. The protein was further purified by gel filtration on Superdex 200.

The synthetic Spir-2-401-427 fragment (QRPRPRVLLKAPTLAEMEEMNTSEEEE) and its N-terminally fluorescein labeled analog: (5-FAM-Ahx-QRPRPRVLLKAPTLAEMEEMNTSEEEE) were purchased from GenScript (Piscataway, NJ, USA).

## Crystallization, data collection, structure determination

The crystallization experiments were performed at 17°C by vapor diffusion in hanging drops. The MyoVa-GTD:Rab11a complex was obtained by mixing the purified MyoVa-GTD and Rab11a in a 1:1

molar ratio to a 12 mg/ml final concentration supplemented with 2 mM beryllium fluoride. Crystals of the MyoVa-GTD:Rab11a complex were grown in 7% (w/v) PEG-8000, 50 mM Bicine, 50 mM Tris, 30 mM NaCl, 20% (v/v) ethylene glycol. The MyoVa-GTD:Spir-2 complex, obtained by mixing MyoVa-GTD with Spir-2-401-427 peptide in 1:3 molar ratio at final protein concentration 20 mg/ml, was crystallized in 50 mM Hepes, 50 mM MOPS, 10% (w/v) PEG 1000, 10% (w/v) PEG 3350, 10% (v/v) MPD. Native data sets were collected to 1.8 Å and 2 Å resolution for MyoVa-GTD:Spir2 complex and MyoVa-GTD:Rab11a complex crystals respectively at Soleil synchrotron PX1 beamline.

The crystals were flash frozen in liquid nitrogen. The X-ray diffraction data were indexed, integrated, and scaled with the XDS program suite (*Kabsch, 2010*). The structures were solved by molecular replacement with Molrep (*Vagin and Teplyakov, 2010*) using MyoVa-GTD structure (PDB ID 4LX1) as a search model. The models were iteratively manually rebuilt with COOT (RRID: SCR_014222) (*Emsley and Cowtan, 2004*), and refined with BUSTER (*Bricogne et al., 2011*) and PHENIX (RRID: SCR_014224) (*Adams et al., 2010*). The data collection and refinement statistics are summarized in *Table 1*.

## Analytical gel filtration

Complex formations were analyzed by analytical gel-filtration using Superdex 200 10/30 Increase column (GE Healthcare Life Sciences) and UV light absorbance at 280 nm and 260 nm and Visible light absorbance at 587 nm at flow rate of 0.75 ml/min in 50 mM Tris pH 8.0, 150 mM NaCl, 2 mM $MgCl_2$, 2 mM TCEP buffer. Fractions of 500 µl were collected and analyzed by SDS-PAGE.

A 100 µl sample of $His_6$-Rab11a-1-173-Q70L at 6 mg/ml, or of MyoVa-GTD at 5 mg/ml or of $His_6$-mCherry-Spir-2-linker at 6 mg/ml were centrifuged at 10,000 x *g* for 10 min and injected into the column to monitor reference elution profiles of the individual proteins. The proteins were mixed pairwise to the same final concentrations and incubated 30 min at 4°C in a total volume of 100 µl. The two-component complexes ($His_6$-mCherry-Spir-2-linker:MyoVa-GTD and MyoVa-GTD:$His_6$-Rab11a-1-173-Q70L) were confirmed by the elution peak shifts and SDS-PAGE analysis. No change in migration of the individual proteins was observed for the $His_6$-mCherry-Spir-2-linker and $His_6$-Rab11a-1-173-Q70L mix. The tripartite $His_6$-mCherry-Spir-2-linker:MyoVa-GTD:$His_6$-Rab11a-1-173-Q70L complex was analyzed at the same concentration of the individual components and confirmed by SDS-PAGE.

$His_6$-mCherry-Spir-2-linker protein was eluted from the gel-filtration column in a single peak but it migrated on SDS-PAGE as three bands with the major band size corresponding to a higher molecular weight than expected (theoretical molecular weight 47 kDa, apparent molecular weight 55 kDa). The purified $His_6$-mCherry-Spir-2-linker protein was analyzed by peptide mass fingerprinting using LC-MS/MS identification approach at Curie Institute Protein Mass Spectrometry Platform. All three bands were identified as $His_6$-mCherry-Spir-2-linker protein or its fragments. The 55 kDa SDS-PAGE $His_6$-mCherry-Spir-2-linker band corresponded to a full-length construct with 65% peptide coverage evenly distributed over the protein sequence, the minor 45 kDa and 12 kDa band species corresponded to $His_6$-mCherry-Spir-2-linker degradation products.

## Cell culture

HEK293 (ATCC Cat# CRL-1573, RRID: CVCL_0045) and HeLa cells (ATCC Cat# CCL-2, RRID: CVCL_0030) were cultured in Dulbecco's Modified Eagle's Medium (DMEM; ThermoFisher) supplemented with 10% (v/v) fetal calf serum (FCSIII; GE Healthcare Life Sciences, HyClone), 2 mM L-glutamine, penicillin (100 units/ml) and streptomycin (100 µg/ml) at 37°C, 5% $CO_2$, 95% humidity and were passaged regularly at 80% confluency. Transfections with plasmid DNA were performed using Lipofectamine reagent (ThermoFisher) according to manufacturer's recommendation.

## Co-immunoprecipitation

HEK293 cells were transfected with expression vectors encoding Myc-tagged Spir proteins and GFP-tagged myosin Va and myosin Vb fragments. 48 hr post transfection, cells were lysed in lysis buffer (25 mM Tris-HCl pH 7.4, 150 mM NaCl, 5 mM $MgCl_2$, 10% (v/v) glycerol, 0.1% (v/v) Nonidet P-40, 1 mM Phenylmethylsulfonylfluorid (PMSF; Sigma-Aldrich), protease inhibitor cocktail (complete mini, EDTA-free; Roche, Penzberg, Germany)) and centrifuged at 20,000 x *g*, 4°C, 20 min to remove insoluble debris. The clarified supernatant was incubated with 4 µg anti-c-Myc antibody (9E10, mouse

monoclonal; Santa Cruz Biotechnology; RRID: AB_627268) for 1 hr on ice. Subsequently, 40 μl Protein G-Agarose beads (1:1 suspension; Roche) were added and incubated for 2 hr at 4°C on a rotating wheel. Beads were washed with pulldown buffer (25 mM Tris-HCl pH 7.4, 150 mM NaCl, 5 mM MgCl$_2$, 10% (v/v) glycerol, 0.1% (v/v) Nonidet P-40). Bound proteins were eluted with 1x Laemmli buffer and denatured for 10 min at 95°C, then analyzed by immunoblotting.

### GST-pulldown from HEK293 lysates

HEK293 cells were co-transfected with expression vectors encoding Myc-Spir-2-FL, mStrawberry-FMN2-FH2-FSI, eGFP-MyoVa-D-FL and eGFP-MyoVa-D-Q1753R, respectively. 24 hr post transfection, cells were lysed in lysis buffer (25 mM Tris-HCl pH 7.4, 150 mM NaCl, 5 mM MgCl$_2$, 10% (v/v) glycerol, 0.1% (v/v) Nonidet P-40, 1 mM PMSF, protease inhibitor cocktail) and centrifuged at 20,000 x $g$, 4°C, 20 min to remove insoluble debris. For GST-pulldown assays 65 μg GST-Rab11a-Q70L protein (25 μg GST control) was coupled to GSH-Sepharose 4B beads (1:1 suspension) for 1 hr, 4°C on a rotating wheel. Beads were washed twice with pulldown buffer (25 mM Tris-HCl pH 7.4, 150 mM NaCl, 5 mM MgCl$_2$, 10% (v/v) glycerol, 0.1% (v/v) Nonidet P-40) and subsequently incubated with the cell lysates for 2 hr at 4°C on a rotating wheel. Beads were washed four times with pulldown buffer and bound proteins were eluted with 1x Laemmli buffer and denatured at 95°C for 10 min, then were analyzed by immunoblotting.

### GST-pulldown from mouse brain

Wild-type C57BL/6 (Jackson Laboratory, Bar Harbor, ME, USA) mice were killed by cervical dislocation. The brain was isolated immediately, washed in cold PBS and lysed in lysis buffer (25 mM Tris-HCl pH 7.4, 150 mM NaCl, 5 mM MgCl$_2$, 10% (v/v) glycerol, 0.1% (v/v) Nonidet P-40, 0.1 M NaF, 1 mM Na$_3$VO$_4$, 1 mM PMSF, protease inhibitor cocktail) using a TissueRuptor (Qiagen). Following incubation on ice for 45 min, the lysate was centrifuged at 20,000 x $g$, 4°C until the supernatant was clear. Brain lysates were incubated with GSH-Sepharose 4B beads, loaded with 50 μg GST-fusion proteins as indicated, for 2.5 hr at 4°C on a rotating wheel. Beads were washed five times with pulldown buffer and bound proteins were eluted by 1x Laemmli buffer and denatured at 95°C for 10 min. Pulled proteins were analyzed by immunoblotting.

### GST-pulldown experiments with purified proteins

For GST-pulldown assays, 50 μg of GST-fusion protein was coupled to GSH-Sepharose 4B beads (1:1 suspension) for 1 hr at 4°C on a rotating wheel. Beads were washed twice with pulldown buffer and subsequently incubated with 20 μg His$_6$-mCherry-fusion peptides for 2 hr, 4°C on a rotating wheel. In order to identify the tripartite Spir-2:MyoVa:Rab11a complex, 65 μg GST-Spir-2-GTBM-SB-FYVE was coupled to beads. Subsequently, beads were incubated with 40 μg MyoVa-GTD and 35 μg Rab11a-Q70L simultaneously for 2 hr. Beads were washed four times with pulldown buffer and bound proteins were eluted by 1x Laemmli buffer and denatured at 95°C for 10 min. Pulled proteins were analyzed by immunoblotting.

### Quantitative GST-pulldown assays

GST-pulldown assays were performed as described above with increasing concentrations of GST-MyoVa/b-GTD fusion proteins and 100 nM His$_6$-mCherry-Spir-2-linker peptide in SPECS buffer (1x PBS, 50 mM NaCl). Beads were pelleted and the supernatant was centrifuged for 10 min at 20,000 x $g$ to remove potential debris that could disturb the fluorescence measurement. Each protein sample was allowed to adapt to 20°C for 15 min in a water bath. The concentration-dependent binding of MyoVa/b-GTD to the Spir-2-linker was determined by fluorospectrometric analysis using FluoroMax-4 Spectrofluorometer (Horiba Jobin Yvon, Bensheim, Germany). The mCherry red fluorescent protein was excited at 548 nm and the emission at 610 nm was recorded (emission maximum). The data were calculated as 'fraction bound' (y) compared to the initial fluorescence signal without any GST-MyoV-GTD protein

$$y = 1 - \frac{y_0 - y_c}{y_0}$$

With $y_0$ is fluorescence signal without GST-MyoV-GTD and $y_c$ is signal at corresponding GST-MyoV-GTD concentration.

Furthermore, data were analyzed in SigmaPlot 12.3 software (Systat Software, Erkrath, Germany). Equilibrium binding data were fitted according to the equation

$$y = \frac{B_{max} \cdot x}{K_d + x}$$

assuming a single binding site and with $B_{max}$ representing the maximal amplitude, $K_d$ representing the equilibrium constant and $x$ representing the concentration of GST-MyoV-GTD.

The binding curves saturated at 40% since the His$_6$-mCherry-Spir-2-linker protein preparation contained C-terminal incomplete protein products, which are still fluorescent but cannot interact with MyoV-GTD. The Spir-2-linker region is predicted to be highly unstructured, which may be the reason for the relative instability of the recombinant His$_6$-mCherry-Spir-2-linker fusion protein.

### Surface plasmon resonance binding assay

The assays were carried out at 25°C in buffer (50 mM Hepes pH 7.5, 100 mM NaCl, 2 mM MgCl$_2$, 1 mM TCEP). 200–250 resonance units (RU; 1RU $\approx$ 1 pg $\times$ mm$^{-2}$) of Rab11a-Q70L(GTP) or Rab11-wt (GDP) were captured through their His$_6$-tag on the surface of a NTA sensorchip using a Biacore 2000 instrument (GE Healthcare Life Sciences). MyoVa-GTD or MyoVb-GTD (10 nM–5 μM) were then injected over the tethered Rabs for one minute at a flow rate of 50 ml/min. The surface was regenerated with a 2-min 0.3 M EDTA incubation and a 1-min 0.1% SDS wash. The real-time interaction profiles were double referenced using the Scrubber 2.0 software (BioLogic Software, Campbell, Australia), both the signals from a reference surface (with a non-relevant protein captured on NTA) and from blank experiments using buffer instead of MyoV-GTD were subtracted. The MyoV-GTD concentration-dependence of the steady-state SPR signals was then analyzed to determine the dissociation equilibrium constants (K$_d$) of the different complexes using the numerical integration algorithm software Biaevaluation 4.1. Each experiment (10 point 2-fold dilution concentration series each) was repeated in triplicate, yielding mean K$_d$ values and standard errors (SEM).

### Fluorescence anisotropy measurements

Fluorescence anisotropy measurements were performed on a PTI Quanta-Master QM4CW spectrofluorometer (PTI, Lawrenceville, NJ, USA) at 25°C using a 10 mm wide quartz cell. Bandwidths of excitation and emission monochromators were set respectively at 5 and 15 nm. Fluorescence anisotropy, expressed in millianisotropy units, was calculated according to the equation

$$A = \frac{IVV - GIVH}{IVV + 2GIVH}$$

with $A$ is anisotropy, $G = IHV/IHH$ is a correction factor for wavelength-dependent distortion, and $I$ is the fluorescence intensity component. Anisotropy values were averaged from 30 different acquisitions. The cell was charged with 1 μM of fluorescein labeled Spir-2-401-427 peptide in a final volume of 1 ml in buffer (100 mM Tris-HCl pH 8.0, 150 mM NaCl, 2 mM MgCl$_2$, 2 mM TCEP, 5% (v/v) glycerol) and titrated with MyoVa-GTD or MyoVb-GTD (50 nM–5 μM). The binding isotherms were fitted to obtain the equilibrium dissociation constants K$_d$ and standard errors, by using the Origin 2015 software that implements Levenberg Marquardt iterative minimization algorithms, using the equation

$$y = F_0 + \frac{F_m - F_0}{2P}\left(P + x + K_d - \sqrt{(P + x + K_d)^2 - 4Px}\right)$$

with $y$ represents the fluorescence anisotropy, $x$ the concentration of MyoV-GTD, $F_0$ the initial anisotropy, $F_m$ the maximum anisotropy, and $P$ the peptide concentration. Experiments were repeated twice with two different protein preparations.

### Microscale thermophoresis measurements (MST)

MST experiments were performed on a Monolith NT.115 system (NanoTemper Technologies, München, Germany) using 5% LED and 30% IR-laser power at 21.5°C. Laser on and off

times were set at 30 s and 5 s, respectively. Fluorescein labeled Spir-2 peptide was synthesized by Genscript, the peptide was dissolved in PBS with 0.005% Tween-20 and diluted to 1 μM concentration. A two-fold dilution series was prepared for the MyoVc-GTD in the interaction buffer (50 mM Tris pH 8.0, 250 mM NaCl, 2 mM MgCl$_2$, 2 mM TCEP, 5% (v/v) glycerol) and each dilution point was transferred to the Fluorescein-Spir-2 solution. The final concentrations of MyoVc-GTD ranged from 97 μM to 2.96 nM, the final concentration of the labeled peptide was 500 nM. Samples were filled into premium capillaries (NanoTemper Technologies) for measurements. The experiments were performed with three independent replicates. The affinity was quantified by analyzing the change in normalized fluorescence as a function of the concentration of the titrated peptide using the NTAnalysis software provided by the manufacturer.

## Immunoblotting

Proteins were separated by SDS-PAGE and analyzed by Western blotting using anti-GFP (Living Colors A.v. peptide antibody, rabbit polyclonal, 1 μg/ml; Clontech; RRID: AB_2313653), anti-RFP (Living Colors DsRed rabbit polyclonal antibody, 0.5 μg/ml; Clontech; RRID: AB_10015246), anti-c-Myc (9E10, mouse monoclonal, 0.4 μg/ml; Santa Cruz Biotechnology; RRID: AB_627268), anti-Spir-1 (SA2133, rabbit polyclonal, 0.5 μg/mL (*Schumacher et al., 2004*); RRID: AB_2619680), anti Rab11 (D4F5 XP, rabbit monoclonal, 1:1000; Cell Signaling Technology, #5589; RRID: AB_10693925) and anti-myosin Va (rabbit polyclonal, 1:750; Cell Signaling Techology, #3402; RRID: AB_2148475) antibodies, horseradish peroxidase linked anti-rabbit IgG (from donkey; RRID: AB_772206) and anti-mouse IgG (from sheep; RRID: AB_772210) secondary antibodies (1:5000, GE Healthcare). Proteins were visualized by chemiluminescence (Luminata Forte Western HRP substrate; Merck Millipore). The signal was recorded with an Image Quant LAS4000 system (GE Healthcare Life Sciences). Recorded images were processed in Adobe Photoshop and assembled in Adobe Illustrator.

## Immunostaining

HeLa cells were seeded on microscope cover glasses and transfected with Myc-epitope-tagged Spir-2 proteins and fluorescently-tagged MyoV and Rab11 proteins as described above. Cells were fixed with paraformaldehyde (3.7% in 1x PBS) for 20 min at 4°C and subsequently permeabilized using 0.2% Triton X-100 (in 1x PBS) for 3.5 min, room temperature. Cells were incubated with anti-c-Myc antibody (9E10, mouse monoclonal, 2 μg/ml, Santa Cruz Biotechnology; RRID: AB_627268) for 1 hr at room temperature and conjugated anti-mouse secondary antibodies (Cy5; from donkey, 3.25 μg/ml, Dianova (RRID: AB_2340820) and TRITC; from donkey, 3.125 μg/ml, Dianova (RRID: AB_2340767)) for 1 hr at room temperature avoiding exposure to light. Finally, cells were mounted on microscope slides with Mowiol and analyzed with a Leica AF6000LX fluorescence microscope, equipped with a Leica HCX PL APO 63x/1.3 GLYC objective and a Leica DFC350 FX digital camera (1392 × 1040 pixels, 6.45 × 6.45 μm pixel size). 3D stacks were recorded and processed with the Leica deconvolution software module. In case of the MyoVa autoregulation/activation experiments, we quantified the MyoVa expression levels. High expression of GFP-MyoVa-D-FL induced vesicular localization, whereas an even cytoplasmic distribution was observed at low expression levels. In order to clearly distinguish between vesicular and cytoplasmic MyoVa localization, we imaged either high or low expressing cells and determined the sum of pixel gray values per cell as a measure for the MyoVa expression levels. Based on that, we only used cells for the autoregulation analysis having a sum of pixel gray values less than $20 \times 10^6$ for the expression of GFP-MyoVa-D-FL. Images were recorded using the Leica LASX software and further processed with Adobe Photoshop and subsequently assembled with Adobe Illustrator.

## Time-correlated single-photon counting fluorescence lifetime microscopy (TCSPC-FLIM)

Fluorescence lifetimes were imaged with a commercial FLIM upgrade kit (PicoQuant, Berlin, Germany) attached to a confocal microscope LSM 880 (Zeiss, Jena, Germany) using a 440 nm pulsed laser diode at 20 MHz repetition rate. The fluorescence was detected using the Big.2-unit (Zeiss) fed into the TimeHarp 260 photon counting board (PicoQuant). Time series of FLIM images were recorded until bright pixels accumulated at least 1000 photon events. Before pixel wise fitting (Symphotime, 64-bit, version 2, PicoQuant), a region of interest was defined manually by setting a

threshold for selecting pixels of high intensity (100–300 photon events per pixel). Fitting employed an exponential two-component model to 20 ns of the decay without binning. Lifetime images contained the color coded, intensity weighted average in each pixel.

For an improved automated selection of vesicular structures in cells, we followed a previously established semi-automated approach (*Austen et al., 2015*). Briefly, confocal imaging was done using a Leica TCS SP5 X confocal microscope equipped with a pulsed white light laser (WLL, 80 MHz repetition rate; NKT Photonics, Birkerod, Denmark), a FLIM X16 TCSPC detector (LaVision Biotec, Bielefeld, Germany) and a 63x water objective (HCX PL APO CS, NA = 1.2). A band-pass filter 514/30 (AHF Analysentechnik, Tübingen, Germany) was used to block acceptor emitted photons. Data analysis includes manual selection of the cytoplasmic regions followed by a threshold algorithm, segmenting high intensity regions based on spatial signal changes at varying levels. The custom-written MATLAB program (*Austen et al., 2015*) calculates the average FRET efficiency $E$ according to

$$E = 1 - \frac{\tau_{DA}}{\tau_D}$$

with $\tau_D$ as the mean donor lifetime $\tau_{DA}$ as mean donor lifetime in presence of an acceptor fluorophore. $E$ represents the entire ROI of a single cell. Here, fitting employed an exponential one-component fit to 10 ns of decay; the first 0.5 ns were omitted due to the instrumental response. Measurements on both instruments, as well as the use of different ROI settings (e.g. cytoplasm with or without threshold excluded regions) reproduced consistent results for three independent experiments.

## Colocalization analysis

The extent of colocalization of Rab11, Spir-2 and MyoVa was analyzed using the ImageJ (V2.0.0) plug-in Coloc 2. Here, the colocalization rate is indicated by the Pearson's Correlation Coefficient (PCC) as a statistical measure to unravel a linear correlation between the intensity of different fluorescent signals. A PCC value of 1 indicates a perfect colocalization, 0 indicates a random colocalization and a PCC value of -1 indicates a mutually exclusive localization of the analyzed signals. To take the noise of each image into account and to gain an objective evaluation of PCC significance, a Costes significance test was performed. Therefore, the pixels in one image were scrambled randomly and the correlation with the other (unscrambled) image was measured. Significance regarding correlation was observed when at least 95% of randomized images show a PCC less than that of the original image, meaning that the probability for the measured correlation of two colors is significantly greater than the correlation of random overlap (*Costes et al., 2004*; *Pompey et al., 2013*). Statistical data analysis was performed using SPSS 22 (IBM, Armonk, NY, USA).

## Accession numbers

The atomic coordinates and structure factors have been deposited in the Protein Data Bank, www.pdb.org, with accession numbers 5JCY (MyoVa-GTD:Spir-2-GTBM), 5JCZ (MyoVa-GTD:Rab11), see *Table 1*.

## Acknowledgements

We acknowledge SOLEIL for provision of synchrotron radiation facilities (proposal 20141015) and we would like to thank Pierre Legrand and Beatriz Guimares for assistance in using beamline PX1. We thank Sandra Lemke, Carsten Grashoff, Yilmaz Niyaz (Zeiss) and Volker Buschmann (Picoquant) for their generous support in FLIM. We thank Mitsuo Ikebe for providing a MyoVb cDNA. We acknowledge Wikayatou Attanda and Charles Gauquelin for support in recombinant protein production. We also acknowledge Damarys Loew and Vanessa Masson (Curie Institute Protein Mass Spectrometry Platform) for the peptide fingerprint analysis of Spir-2 constructs. BG, AHo, OP, GM, FC are part of the Labex CelTisPhyBio 11-LBX-0038, which is part of the IDEX PSL (ANR-10-IDEX-0001-02 PSL).

# Additional information

## Funding

| Funder | Grant reference number | Author |
|---|---|---|
| Centre National de la Recherche Scientifique | | Anne Houdusse<br>Olena Pylypenko |
| Deutsche Forschungsgemeinschaft | SPP 1464: KO 2251/13-1 | Martin Kollmar |
| National Science Foundation | MCB-1244235 | Margaret A Titus |
| University of Minnesota | Medical School Award, E-0915-04 | Margaret A Titus |
| Deutsche Forschungsgemeinschaft | SPP 1464: SCHW 716/11-1, -2 | Petra Schwille |
| Ligue Contre le Cancer | RS16/75-67 | Anne Houdusse |
| Fondation ARC pour la Recherche sur le Cancer | SFI20121205398 | Anne Houdusse |
| Deutsche Forschungsgemeinschaft | SPP 1464: KE 447/10-1, -2 | Eugen Kerkhoff |
| Bayerische Forschungsstiftung | WFKMS Nr. F2 - 5121.4.3.1 -10c (BayGene) | Eugen Kerkhoff |
| Fondation de la Recherche Medicale | ING20140129255 | Anne Houdusse |
| Agence Nationale de la Recherche | ANR-13-BSV8-0019-01 | Anne Houdusse |

The funders had no role in study design, data collection and interpretation, or the decision to submit the work for publication.

## Author contributions

OP, Designed research, Determined the structures, Measured direct binding interactions, Acquisition of data, Analyzed data, Wrote the paper; TWel, Performed biochemical and cell assays to characterize MyoV:Spir and Spir:MyoV:Rab11 complexes, Purified proteins for binding experiments, Acquisition of data, Analyzed data, Wrote the paper; JT, TWei, Performed and analyzed FLIM-FRET cell colocalization experiments, Acquisition of data, Analysis and interpretation of data; MK, Performed a bioinformatics analysis of the Spir and MLPH myosin binding domains, Acquisition of data, Analysis and interpretation of data; FC, GM, Purified proteins for structural and binding experiments, Acquisition of data; SW, CILM, Helped perform biochemical and cell assays to characterize MyoV:Spir and Spir:MyoV:Rab11 complexes, Acquisition of data, Analysis and interpretation of data; AS-W, ATG, Helped perform biochemical and cell assays to characterize MyoV:Spir and Spir:MyoV: Rab11 complexes, Acquisition of data; AHu, BG, Provided reagents and contributed in the experimental design and discussion; BB, PE, Measured direct binding interactions, Acquisition of data, Analysis and interpretation of data; MAT, Analyzed data, Wrote the paper; PS, Performed and analyzed FLIM-FRET cell colocalization experiments, Acquisition of data; AHo, Designed research, Analyzed data, Wrote the paper; EK, Designed research, Acquisition of data, Analyzed data, Wrote the paper

## Author ORCIDs

Andreas Till Grasskamp, http://orcid.org/0000-0002-5895-6529
Petra Schwille, http://orcid.org/0000-0002-6106-4847
Eugen Kerkhoff, http://orcid.org/0000-0002-3640-7916

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
