## [Decision Letter]

Thank you for submitting your article "Coordinated recruitment of Spir actin nucleators and myosin V motors to Rab11 vesicle membranes" for consideration by *eLife*. Your article has been favorably evaluated by Randy Schekman (Senior editor) and three reviewers, one of whom, Pekka Lappalainen (Reviewer #1), is a member of our Board of Reviewing Editors. The following individuals involved in review of your submission have agreed to reveal their identity: Roberto Dominguez (Reviewer #2); Giorgio Scita (Reviewer #3).

The reviewers have discussed the reviews with one another and the Reviewing Editor has drafted this decision to help you prepare a revised submission.

Summary:

The mechanisms by which myosin motor proteins and actin filament nucleation machineries are coordinated in various cellular processes are incompletely understood. Here, Pylypenko et al. approached this question by studying how the actin filament nucleator Spir is targeted to myosin V/Rab11 containing vesicles. Using biochemical and cell biological approaches they revealed that a conserved sequence motif in Spir's central linker region binds MyoVa and MyoVb, and they then determined the structural mechanism of this interaction. A curiosity of the structure is that it shows partial overlap between Spir's GTD-Binding Motif and the binding site of melanophilin (MLPH) on the Globular Tail Domain (GTD) of myosin V. They also provide evidence that this interaction is necessary to target Spir to Rab11-positive vesicles, and that it contributes to activation of MyoV. Finally, the authors present the crystal structure of MyoVa's GTD in complex with Rab11.

This manuscript provides valuable new information about the interplay between Spir-dependent actin filament assembly, myosin V motors, and the Rab11 GTPase at membranes. However, there are several issues that should be addressed to strengthen the manuscript.

Essential revisions:

1) All three reviewers felt that the paper would be stronger if it included evidence about the functional role of these interactions in cells. The reviewers strongly suggest that the authors provide such evidence. For instance, a Spir-2 mutant no longer capable of binding MyoVa/b (such as the LALA mutant of Spir-2 shown in Figure 1) could be exploited in knockdown-rescue experiments to test the physiological relevance for the identified trimeric Spir-MyoV-Rab11 complex. This could perhaps be approached by using the system described in Morel et al., Dev Cell (2009), where Spir-1 knockdown in HeLa cells results in a decreased number of endocytic actin patches and diminished Lamp1-dextran co-localization. In the event that such experiments turn out to be unfeasible within the timeframe allowed for resubmission by *eLife*, the authors should at the minimum discuss this issue (the physiological role of the interaction) and propose follow-up experiments to address this matter.

2) The fact that Spir and Rab11 co-localize has been known for many years (Kerkhoff et al., Curr. Biol 2001). Furthermore, the structure of MyoVb's GTD in complex with Rab11 was published by the authors (Pylypenko et al., 2013). The new structure shown here of MyoVa's GTD in complex with Rab11 is a little different, but not 'dramatically' different as stated in the manuscript. For these reasons, the very long section 'The MyoVa-GTD/Rab11 interaction' is a distraction here, diluting the main message. A simple mention of this new structure would be sufficient. Accordingly, Figure 5 should be moved to supplemental data. Furthermore, the following section 'A tripartite[…]' can also be shortened, since it is confirmatory of facts that are either published or obvious. As a result, the paper will be better – shorter and more focused.

3) Overall, the paper would be far better (more effective and focused) if matters of discussion are moved to the 'Discussion' section and the very long introductions at the beginning of each section are kept to the point. Further extending this article is unnecessary, since the major points are important enough, and a shorter focused paper is likely to be better received.

---

## [Author Response]

*Essential revisions:*

1) All three reviewers felt that the paper would be stronger if it included evidence about the functional role of these interactions in cells. The reviewers strongly suggest that the authors provide such evidence. For instance, a Spir-2 mutant no longer capable of binding MyoVa/b (such as the LALA mutant of Spir-2 shown in Figure 1) could be exploited in knockdown-rescue experiments to test the physiological relevance for the identified trimeric Spir-MyoV-Rab11 complex. This could perhaps be approached by using the system described in Morel et al., Dev Cell (2009), where Spir-1 knockdown in HeLa cells results in a decreased number of endocytic actin patches and diminished Lamp1-dextran co-localization. In the event that such experiments turn out to be unfeasible within the timeframe allowed for resubmission by eLife, the authors should at the minimum discuss this issue (the physiological role of the interaction) and propose follow-up experiments to address this matter.

We very much appreciate the wish to fully understand the cell biological functions of the protein interaction network characterized in our study. Cell biological studies of Spir function have been complicated in the past by the relative low expression of the proteins. Transient overexpression of Spir proteins (wt and mutants) severely alters the morphology and motility of Rab11 vesicles (Figure 5, mStrawberry Rab11). This makes rescue experiments of knockdown/knockout phenotypes very difficult, especially if they require a quantitative analysis. The novel CRISPR/Cas9 technologies today allow targeted genome editing in immortalized cell lines and thereby the study of proteins and mutants at their endogenous expression levels. In our opinion this will be necessary to quantitatively address the cell biological functions and uncover the complicated equilibrium of protein interactions that govern the actin and microtubule dependent transport of Rab11 vesicles.

Since these experiments are not feasible within the timeframe allowed for resubmission by eLife, we have added a discussion of the function of the Spir/MyoV interaction in Rab11 vesicle transport processes and proposed experiments to address this, as suggested by the reviewers.

2) The fact that Spir and Rab11 co-localize has been known for many years (Kerkhoff et al., Curr. Biol 2001). Furthermore, the structure of MyoVb's GTD in complex with Rab11 was published by the authors (Pylypenko et al., 2013). The new structure shown here of MyoVa's GTD in complex with Rab11 is a little different, but not 'dramatically' different as stated in the manuscript. For these reasons, the very long section 'The MyoVa-GTD/Rab11 interaction' is a distraction here, diluting the main message. A simple mention of this new structure would be sufficient. Accordingly, Figure 5 should be moved to supplemental data. Furthermore, the following section 'A tripartite[…]' can also be shortened, since it is confirmatory of facts that are either published or obvious. As a result, the paper will be better – shorter and more focused.

To make the paper more focused on the new interaction between Spir and MyoV, the MyoVa:Rab11 structure description in the main text was significantly shortened and merged with the shortened “A tripartite complex[…]” section. We also moved the figure describing details of the MyoVa:Rab11 structure to the supplemental data.

*3) Overall, the paper would be far better (more effective and focused) if matters of discussion are moved to the 'Discussion' section and the very long introductions at the beginning of each section are kept to the point. Further extending this article is unnecessary, since the major points are important enough, and a shorter focused paper is likely to be better received.*

We followed the reviewer’s recommendation and minimized the introductory information at the beginning of each section and moved discussion relevant information in the 'Discussion' part.